# Immunohistochemical and Ultrastructural Analysis of Adult Neurogenesis Involving Glial and Non-Glial Progenitors in the Cerebellum of Juvenile Chum Salmon *Oncorhynchus keta*

**DOI:** 10.3390/ijms26199267

**Published:** 2025-09-23

**Authors:** Evgeniya V. Pushchina, Mariya E. Bykova, Evgeniya E. Vekhova, Evgeniya A. Pimenova

**Affiliations:** A.V. Zhirmunsky National Scientific Center of Marine Biology, Far Eastern Branch, Russian Academy of Sciences, 690041 Vladivostok, Russia

**Keywords:** chum salmon *Oncorhynchus keta*, cerebellum, adult neurogenesis, adult-type glial progenitors, adult-type non-glial progenitors, neuroepithelial cells, eurydendroid neurons, mossy fibers, climbing fibers, dorsal matrix zone, PCNA, GFAP, HuCD, vimentin, nestin

## Abstract

The ultrastructural organization of different cell types involved in homeostatic growth in the cerebellum of juvenile chum salmon (*Oncorhynchus keta*) was investigated using transmission and scanning electron microscopy. The organization of astrocytes, oligodendrocytes, dark cells, adult-type glial and non-glial progenitors, stellate neurons, and eurydendroid cells (EDCs) in the molecular and granular layers and granular eminences was characterized. The organization of dendritic bouquets of Purkinje cells and climbing fibers was studied for the first time at the ultrastructural level, and the ultrastructural features of mossy fibers and the rosettes they form were characterized. Scanning electron microscopy (SEM) revealed the presence of single and paired adult-type neural stem/progenitor cells (aNSPCs) on the cerebellar surface and stromal clusters of aNSPCs outside the dorsal matrix zone (DMZ). Immunohistochemical (IHC) verification of proliferating cell nuclear antigen (PCNA) revealed five types of proliferating cells in the cerebellum of juvenile chum salmon: neuroepithelial cells (NECs), glial aNSPCs, and non-glial aNSPCs. A glial fibrillary acidic protein-positive (GFAP) complex consisting of radial glial fibers and aNSPCs was detected in the DMZ. At the same time, a complex of GFAP+ cerebellar afferents, consisting of differentiating mossy and climbing fibers, was found to develop in the cerebellum of juvenile chum salmon. Nestin+ non-glial aNSPCs and small nestin+ resident cells were detected in the dorsal, lateral, and basal areas, as well as in the granular layer (GrL) and granular eminences (GrEm). These cell types may contribute to the homeostatic growth of the cerebellum by acting as both active participants (PCNA+) and resident (silent) aNSPCs. Studying vimentin-positive systems in the cerebellum revealed a widespread presence of proliferating glial aNSPCs that actively contribute to homeostatic growth, as well as small resident immunopositive cells throughout the cerebellum of juvenile chum salmon. Immunolocalization of the neuronal RNA-binding protein marker (HuCD) was detected in numerous molecular layer (ML) cells at the early stages of neuronal differentiation in the dorsal and lateral regions of the cerebellum of juvenile chum salmon. HuCD + EDCs were detected for the first time in the dorsal (DZ) and basal (BZ) zones, forming broad axonal arborization. Immunolabeling of HuCD in combination with transmission electron microscopy (TEM) allowed EDCs to be characterized in the cerebellum of juvenile chum salmon for the first time.

## 1. Introduction

It is widely accepted that adult neurogenesis in different species is linked to regenerative capacity, learning, and memory, including spatial, contextual, and emotional memory [1]. Numerous comparative studies on various vertebrate species have shown that neurogenic abilities in adults are limited to the anterior regions of the brain throughout vertebrate evolution [2]. Fish exhibit the greatest neurogenic potential, associated with ongoing brain growth and substantial regenerative capacity [3,4]. Most studies on fish have focused on teleosts, particularly the zebrafish *Danio rerio* [5], the knifefish *Apteronotus leptorhynchus* [6], and the three-spined stickleback *Gasterosteus aculeatus* [7], and cartilaginous fish [8], including the stingray *Raja asterias* and the electric ray *Torpedo ocellata* [9].

Fish exhibit widespread post-embryonic neurogenesis originating from numerous proliferative niches distributed along the brain axis [5,10,11]. The cerebellum of salmonids is a suprasegmental center with neurogenic activity, as are other caudal segmental centers of the brainstem [12]. During central nervous system development, various cell types appear in a specific temporal sequence from increasingly polarized precursors. However, it remains unclear whether neural stem cells and progenitor cells with limited potential, or stem cells with greater potential, are present in the brains of bony fish.

Currently, it is generally accepted that adult progenitor cells in the mammalian brain can be divided into radial glia-like and non-glial progenitor cells [13,14]. Radial glia-like progenitor cells have the ability to self-renew and maintain a long-term undifferentiated state, generating various types of neurons [14]. However, different teleost species exhibit phenotypic features of the aNSPCs pool. In particular, radial glia (RG) cells corresponding to glial aNSPCs have not been identified in juvenile masu salmon (*Oncorhynchus masou*) [15] or chum salmon (*O. keta*) [16]. Instead of the typical RG cells, *O. masou* and *O. mykiss* trout have various astrocyte-like cells that express vimentin, nestin, and glutamine synthetase (GS) [15,16]. During the post-traumatic period, GS+ RG cells have been identified in juvenile *O. masou*, but no GS+ RG cells have been detected in the cerebellum of intact *O. masou* juveniles [15]. Studies of the ultrastructural organization of the tegmental area of juvenile chum salmon (*O. keta*) showed the presence of an astrocyte population in the neurogenic zones of the *tegmentum* and *torus semicircularis* [17].

Studies of mammals have shown that glial progenitor cells (radial glia) can divide to generate non-glial progenitors [13,18]. However, these cells are usually in a silent state. In the mammalian brain, they are known as B cells in the subventricular zone (SVZ) and type 1 cells in the subgranular zone (SGZ) [14,19,20]. These cells express GFAP, brain lipid-binding protein (BLBP), GS, and Sox2 [21]. Cells expressing GFAP, BLBP, GS, and nestin have been identified in the cerebellum of *O. masou* [15] and *O. keta* [16,17].

Another population of cells in the mammalian brain, non-glial progenitor cells, are transit-amplifying cells [22] or intermediate progenitor cells (IPCs). These actively proliferating cells lack radial processes are positive for proliferation markers (PCNA and Ki67), incorporate bromodeoxyuridine (BrdU) during S-phase, and express neural lineage markers (HuCD and neurogenin). Their future phenotype depends on these markers [23,24]. Studies on juvenile chum salmon *O. keta* have shown the existence of several alternative adult-type precursors in the cerebellum, as well as neuroepithelial-type cells [17,25]. However, the immunohistochemical profile of these cells is not known.

In mammals, IPCs undergo mitosis, resulting in the formation of either several IPCs or two migrating neuroblasts. These neuroblasts then leave the neurogenic niche and migrate to their final destination in the brain. In the SVZ of mammals, these neuroblasts are called type A cells and migrate along a specific tangential path to the olfactory bulb, known as the rostral migratory stream (RMS) [20]. In the SGZ, these cells are called type 3 cells, locally migrating to their final destination in the hippocampus. In mammals, neuroblasts in SVZ and SGZ express the same markers of origin as IPCs. In salmon studies, mammalian IPC-like cells have been identified in various regions of the brain, including the telencephalon [25], the cerebellum [15,16], and the mesencephalic tegmentum [17]. Nevertheless, further research is required to confirm the status and biology of IPC-like cells in juvenile salmon. This study aimed to examine the ultrastructural features of aNSPCs in the cerebellum of juvenile chum salmon and determine their immunohistochemical profile.

## 2. Results

### 2.1. Ultrastructural Organization of the Cerebellum of Juvenile Chum Salmon

#### 2.1.1. The Molecular Layer

In the ML of the cerebellum of juvenile chum salmon, stellate neurons, dendritic bouquets of Purkinje cells (PCs), oligodendrocytes, astrocytes, and myelinated fibers were identified (Figure 1A–F). The ML of the juvenile chum salmon cerebellum contains stellate cells (SCs), which are located in the upper third and deeper layers (Figure 1A). These SCs resemble spheroidal microneurons with short dendrites. The soma size of SCs is shown in Table 1. A large nucleus containing lumpy heterochromatin was located in the basal part of the cell (Figure 1A, Table 1). The cytoplasm of the SCs was of medium density with elongated large and small mitochondria, which were evenly distributed (Figure 1A, purple triangular arrows). Numerous myelinated fibers forming the fibrous layer were detected in the lower third of the ML (Figure 1B). The ends of climbing fibers (cf) and parallel fibers were found in the same area (Figure 1B). Oligodendrocytes forming myelin complexes on the surface of smooth fibers were found in the infraganglionic plexus area (Figure 1B).

Data on trout [26] show that axonal and dendritic branches of Purkinje cells appear quite early in salmonids. It has been demonstrated that the axon and its collaterals are more advanced in development than the dendritic tree of PCs. In juvenile chum salmon, growth cones and filopodia were found on PC dendrites (see Figure 1C), suggesting that PC dendrites grow and form new branches at these sites. PC dendrite fragments at these ML levels have light cytoplasm, a large number of mitochondria of various morphologies (Figure 1C, yellow triangular arrows), and vacuole-like cytoplasmic inclusions (Figure 1C). Adjacent to the PCs dendrites are cf with an intricate and convoluted morphology (Figure 1C). The terminal branches of the cf spread in parallel with the dendritic branches of the PCs, following a contour path and wrapping around the dendritic processes (Figure 1C). Oligodendrocytes forming myelin sheaths on the surface of the cf were identified next to the dendritic bouquets of the PCs (Figure 1C,D). Oligodendrocytes occupied a fairly large space in the ML and were involved in the myelination of the fibrous layer (see Figure 1D). Filopodia were also found at the tertiary and higher-order branches of PC dendrites in juvenile chum salmon (Figure 1C), consistent with data on mature trout. In juvenile chum salmon, the dendritic bouquets of PCs are often covered with spikes that represent sites of synaptic contact (Figure 1D,E). The density of the synaptic environment of PC dendrites may differ in different areas (Figure 1F), but the patterns of dendritic structures are generally quite typical.

Protoplasmic-type astrocytes were detected on the surface of EDCs (Figure 2A). Studies on trout have shown that PCs and EDCs can be distinguished based on their afferent synapses [27]. In trout fry measuring 20–30 mm, the surface of PCs and EDCs is uneven due to the presence of perisomatic processes. Similar structures have been described for developing PCs in birds and mammals [28,29,30], where they are more numerous. Perisomatic processes are usually wider than filopodia but contain a similar amount of thin filamentous material. In some cases, however, they contain ordinary cytoplasmic organelles. At later stages of development, the surface becomes smoother due to the growth of the entire cell. According to this theory, the presence of perisomatic processes on cells is associated with growth, which is consistent with Mugnaini’s findings [30]. The dendrites of PCs in trout exhibit an ultrastructure that has been observed in higher vertebrates.

The proximal EDC dendrites of juvenile *O. keta* differ from those of the PCs because their cytoplasm is less electron dense, subsurface cisterns are absent or very rare, and they contain fewer microtubules which are less regularly arranged (see Figure 2B). The distal EDC dendrites bear only a few small spines. To date, no attempts have been made to distinguish these processes from other dendrites with a small number of spines, such as those of Golgi cells and SCs.

Protoplasmic-type astrocytes with few appendages have been identified next to the EDCs (Figure 2B). Previously, contacts between astrocytes and PCs were detected in the cerebellum of juvenile chum salmon (*O. keta*) [31], as well as between astrocytes and large tegmental neurons [17]. The neuro-glial relationship of projection neurons, PCs and EDCs, within the ganglion layer of salmonids indicates a high degree of GL differentiation.

The aim of this study was to characterize the ultrastructure and immunohistochemistry of adult glial and non-glial precursors. In the ML of juvenile chum salmon, we identified several types of cells that appear to be derived from glia. The first type was represented by the so-called “dark cells” previously identified in the trout cerebellum [26]. Similar cells were detected in the ML of juvenile chum salmon (Figure 2C). Dark cells were found next to SCs in the ML. Their sizes were 5.4 ± 0.6/3.2 ± 0.7 µm (Figure 2D). The dark color of these cells is due to their high ribosome content. Glia-like cells of a different phenotype (type II) in the ML of juvenile chum salmon had an irregularly shaped nucleus containing heterochromatin, surrounded by a middle layer of homogeneous cytoplasm. These cells measured 5.7 ± 0.5/4.2 ± 0.6 µm (Figure 2E) and sometimes coexisted with developing EDCs (Figure 2E). Finally, type III was represented by small cells measuring 5.5 ± 0.6/3.8 ± 0.7 µm with an irregularly shaped nucleus containing lumpy chromatin and a narrow layer of dark, dense cytoplasm (Figure 2F). Determining the immunohistochemical profile of these cells may help clarify their involvement in homeostatic growth and repair processes.

#### 2.1.2. Ultrastructural Organization of the Dorsal Matrix Zone

The dorsal matrix zone (DMZ) of the cerebellum of juvenile chum salmon is the main neurogenic region in adult animals (Figure 3A). The heterogeneity of the cellular composition of DMZ, previously presented on semithin sections [31], showed the presence of elongated neuroepithelial cells (NECs) on the periphery (Figure 3B). Ultrastructural analysis data confirmed the presence of individual NEC located in the peripheral parts of the DMZ (Figure 3B). Another cell type is represented by rounded, light-colored cells (Figure 3A, inset), with soma dimensions of 7.71 ± 0.45 by 6.06 ± 0.53 µm and a centrally positioned round nucleus measuring 6.65 ± 0.36 by 5.46 ± 0.54 µm (Table 1). We classify these cells as adult-type non-glial progenitors (aNSPCs), which possess light cytoplasm and a pale nucleus containing reticular euchromatin (Figure 3C). Such cells were detected in multiple cerebellar regions, including the dorsal DMZ (Figure 3A,C), the ventrolateral zones, and the surface layers of the dorsal and dorsolateral regions.

The third cell type consists of numerous dark-colored, stellate-shaped adult-type glial aNSPCs occupying the central region of the DMZ (Figure 3A–C). This population is heterogeneous and comprises both small cells and larger precursor types (Table 1). According to the classification by Lindsey [32], these cells correspond to type III glial progenitors, characterized by an electron-dense cytoplasm, a large irregularly shaped nucleus, and short processes (Figure 3B). Type III cells form clusters in the peripheral regions of the DMZ (Figure 3C), where they are positioned adjacent to young neurons and/or neuroblasts. Large, elongated cells with bipolar morphology, corresponding to pericytes or astrocytes (Figure 3D), were also detected in the DMZ and were surrounded by numerous myelinated fibers.

Analysis of the distribution of adult-type glial and non-glial precursors in the ventrolateral part of the cerebellum showed that non-glial precursors form large clusters in the parenchyma of the ML (Figure 3E), as well as along the ML border (Figure 3E). In the GrL, non-glial aNSPCs may alternate with clusters of type III glial aNSPCs (Figure 3F), where intercellular patterns of secretory granule and myelinated fiber localization have also been observed (Figure 3F). In general, the TEM results combined with the SEM data suggest that two types of precursors are present in the superficial areas of the cerebellum of juvenile chum salmon, including the granular eminences. The first type consists of rounded or spherical cells forming clusters within the ML parenchyma and on the surface of the dorsal cerebellar zone, corresponding to non-glial precursors (aNSPCs).

Another type of precursor, characterized by an irregularly shaped nucleus with uneven edges and located within the DMZ parenchyma and the GrL of the cerebellar body, corresponds to glial aNSPCs. Thus, TEM and SEM analyses demonstrated the presence of two precursor types involved in the homeostatic growth of the cerebellum in juvenile chum salmon.

#### 2.1.3. Granular Layer

The granular layer is the deepest layer of the cerebellum, located basal to the basal granular zone (GrL). Several types of granule cells, as well as Golgi cells [31], were identified in the GrL of the juvenile chum salmon cerebellum (Figure 4A–F). The development of granule cells in juvenile chum salmon displays greater diversity compared to higher vertebrates. The axon of these cells may form a T-shaped or unbranched process, depending on the migration pathways of their progenitors. Ultrastructural analysis of the cerebellar GrL in juvenile chum salmon revealed differences in the morphological structure of granule cells, forming a heterogeneous group of adult-type precursors, as well as glutamatergic neurons that extend parallel fibers branching in a T-shaped manner within the ML.

#### 2.1.4. Granule Cells

In the dorsomedial part of the cerebellum, the neurons of the GrL are represented by small cells measuring of 6.19 ± 0.71 µm and 3.96 ± 0.53 µm (Figure 4A). Granule cells (GrCs) possess large, rounded or oval light nuclei containing constitutive clumped heterochromatin, which is also distributed along the nuclear periphery (Figure 4A). The cytoplasm of GrCs is typically sparse, light, and foamy, containing vacuoles and mitochondria (MT) of varying sizes, and surrounds the nucleus (Figure 4A). Cells with electron-dense, homogeneous nuclei of irregular shape and a narrow rim of dark cytoplasm devoid of organelles were classified as type III aNSPCs (Figure 4A), according to the generally accepted Lindsey classification [32]. Among the GrCs, more elongated cells (5.67 ± 1.13 µm) and smaller cells (3.81 ± 0.42 µm) with an elongated, irregularly shaped nucleus, finely clumped heterochromatin, and a narrow rim of cytoplasm devoid of organelles were also observed (Figure 4A). These cells correspond to type IV aNSPCs according to the Lindsey classification (Figure 4B) [32].

Another population of cells with an oval or ellipsoid nucleus measuring 7.7 ± 0.45/6 ± 0.53 µm with reticulated euchromatin was also identified (Figure 4A, shown in light green). We consider this cell population as a transitional form between a non-glial progenitor cell and a neuroblast (Figure 4A). Along with GrCs and precursors of types III and IV, myelinated and non-myelinated fibers were found in the GrL (Figure 4A,C).

In the basolateral region, the ratio of GrCs to type IV precursors varies and is approximately 1:1 (Figure 4C). In this cerebellar region, type IV aNSPCs tend to form clusters, with small aggregates appearing and the diffuse distribution pattern shifting toward a clustered arrangement (Figure 4C). In the lateral part of the cerebellar body within the GrL, mixed diffuse distribution patterns of GrCs and type III and IV precursors were observed, without cluster formation (Figure 4D). In the mediobasal zone, similar clusters formed by type III aNSPCs were identified, which we define as constitutive adult-type neurogenic niches (Figure 4E). Interestingly, branches and terminals of cf were observed adjacent to the neurogenic cluster (Figure 4E). Figure 4F presents an adult-type neurogenic niche formed by type III aNSPCs at higher magnification.

### 2.2. Ultrastructural Organization of Cerebellar Cells (SEM)

The potential of SEM to detect the surface morphology of nervous tissue has been successfully used in the central nervous system (CNS), with special emphasis on the ventricular surface [33,34] and the superficial subpial regions [31]. However, relatively few SEM studies on cerebellar cytoarchitecture have been published to date. In our laboratory, SEM was used to analyze the organization of the cerebellum of juvenile chum salmon under conditions of homeostatic growth. According to previous studies, the cerebellar cortex of fish consists of three layers: GrL, ganglion layer (GL), and ML, which are homologous to the layers of higher vertebrates. A fibrous plate (infraganglionic plexus) formed by axons of PCs has been described between GrL and GL [35]. The aim of this section is to examine the superficial cytoarchitecture of the juvenile chum salmon cerebellum using SEM, focusing on the spatial distribution of aNSPCs in superficial regions and on mapping intracerebellar neural circuits at the GrL level.

Spherical cells identified in the surface dorsolateral regions of the ML were selected for stereoscopic ultrastructural examination. We previously demonstrated that in trout *O. mykiss* [12], masu salmon *O. masou* [36] and chum salmon *O. keta* [37], such cells are PCNA+ and also express vimentin and nestin [12,15]; thus, they are adult-type precursors. SEM was used to detail their spatial ultrastructure. The results of the study showed that these cells have a heterogeneous surface structure, with perisomatic thickenings, anchored by microfibrils on the surface of the ML (Figure 5A). Due to aNSPC proliferation, paired clusters were often formed (Figure 5B). Large stromal clusters of aNSPCs were observed in the dorsolateral regions (Figure 5C). In the area of the granular eminence, diffuse aNSPC distribution patterns often showed their association with the surface matrix (Figure 5D). Surrounded by components of the pial extracellular matrix, aNSPCs were often found next to biconcave erythrocytes with a smooth surface (Figure 5E,F). Overall, SEM data suggest that spherical cells with superficial localization in the cerebellar body can be regarded as adult-type precursors.

### 2.3. Cells Proliferation in the Cerebellum of Juvenile Chum Salmon. Immunolabeling of Proliferating Cell Nuclear Antigen (PCNA)

Proliferative activity in the cerebellum of juvenile chum salmon was observed in almost all areas of the ML (Figure 6A–P). The morphometric parameters of PCNA+ cells and their immunolabeling intensity are presented in Table 2.

Numerous PCNA+ cell populations with different localizations in the ML were identified in the dorsal and dorsolateral regions of the cerebellum (Figure 6A). Oval and round intensely labeled PCNA+ cells were identified in the surface subpial layer (Figure 6B, Appendix A). The number of oval cells measuring 7–7.5 µm was 132, while the number of rounded cells measuring 4.5–5 µm was 94 (Figure 6Q). The third type of cell was represented by a surface-migrating population with moderate and/or intense PCNA labeling (Figure 6B). The number of such cells, measuring 2–5 µm, was low. In the deeper superficial layers of the ML, tangential migration patterns of elongated PCNA cells measuring 7–8 µm were observed (Figure 6B,C).

Another type of migrating PCNA+ cells, measuring 10–11 µm with moderate immunolabeling, was identified as radially migrating (Figure 6B–D). Radial and tangential migration patterns in the dorsal and dorsolateral regions of the cerebellar body of juvenile chum salmon were quite typical and occurred at different levels of the cerebellar body in the rostrocaudal direction. Occasionally, individual, intensely immunolabeled PCNA cells of types 1 and 2 were observed in the upper third of the ML, as well as along radial migration guides (Figure 6D). These cells were predominantly solitary and did not form cell clusters (Figure 6D). In the lateral region of the cerebellar body, the number of PCNA cells of types 1 and 2 was reduced, whereas the tangential and radial migration patterns of PCNA+ cells of types 3 and 4 remained distinct (Figure 6E). Among the radially migrating cells, both intensely and moderately immunolabeled PCNA cells were identified (Figure 6E).

Intensely PCNA+ cells of types 1 and 2, as well as elongated intensely labeled PCNA+ clusters of NECs, were observed in the dorsal matrix zone (DMZ) (Figure 6F). In the dorsal part of the DMZ, PCNA+ cells of types 1 and 2 formed small, superficially located clusters (Figure 6G, Appendix A). NECs were located ventrally, forming dense, intensely immunolabeled PCNA+ clusters or were located separately (Figure 6G, Appendix A). Patterns of radial migration of moderately or intensely labeled PCNA cells were observed near the DMZ (Figure 6G, Appendix A). In the ventral part of the DMZ, intensely PCNA+ cells of types 1 and 2 formed a local cluster (Figure 6H).

In the lateral part of the cerebellar body, individual PCNA+ cells of types 1 and 2 located in the parenchyma, as well as clusters of small PCNA+ cells in the lower third of the ML, were observed in the deeper layers of the ML (Figure 6I, Appendix A). Patterns of radial migration of smaller PCNA+ cells, measuring 1–3 µm, were observed in this area (Figure 6I, Appendix A). In the deeper layers, individual type 1 PCNA+ cells were identified in the GL and GrL, while more numerous single type 2 PCNA+ cells were detected in the lower third of the ML (Figure 6J).

An extensive population of PCNA+ cells was observed in the basal matrix zone (BMZ) of the cerebellar body (Figure 6K). Being a derivative of the cerebellar ventricle, the BMZ forms a ventral cluster of PCNA+ cells while maintaining a residual ventricular cavity (Figure 6K). Another cluster of intensely immunolabeled PCNA+ cells of types 1 and 2 was detected in the subventricular zone bordering the dorsal part of the IV ventricle of the brain (Figure 6K). Separate intensely labeled PCNA+ cells of types 1 and 2, as well as small clusters of such cells, were identified in the BMZ (Figure 6L). A large accumulation of PCNA+ and PCNA-negative cells was detected on the ventral wall of the cerebellum, bordering the dorsal part of the IV ventricle of the brain (Figure 6M). Various cell types were identified in the heterogeneous cluster, including numerous PCNA-negative cells in the dorsal part of the cluster, and PCNA+ cells of types 1 and 2, as well as small clusters of such cells in various areas of the cluster (Figure 6N). In the ventrolateral regions of the ventral wall of the cerebellum bordering the IV ventricle of the brain, separate intensely labeled PCNA+ cells of types 1 and 2 along with their clusters in the PVZ and SVZ were identified (Figure 6O). In the ventromedial zone, along the central axis, the density of such cell groups in the PVZ was increased (Figure 6P).

A quantitative analysis of the distribution of PCNA+ cells in various regions of the cerebellum showed that the majority of proliferating precursors were localized in the GrL and GrEm (Figure 6Q, Table 2). In the dorsal and lateral zones, precursors of all five types were identified, including type 1 and type 2 cells corresponding to adult-type glial precursors, type 3 cells corresponding to non-glial precursors, type 4 cells corresponding to migrating juvenile cells, and type 5 cells corresponding to NECs.

### 2.4. Localization of Glial Fibrillary Acidic Protein (GFAP) in the Cerebellum of Juvenile Chum Salmon

A study of GFAP immunolocalization in the cerebellum of juvenile chum salmon showed that most of the labeled structures were afferents, mossy fibers (mf) and climbing fibers (cf). Other structures, apparently, represented Bergmann fibers of radial glia (Figure 7A–D), with the highest distribution density observed in the DMZ. A small cell population consisted of GFAP+ cells measuring 4, 5 and 6 µm (large diameter), which, unlike the fibers, had a limited distribution in the cerebellum of juvenile chum salmon. The morphometric characteristics of GFAP+ cells are presented in Table 3.

The largest neurogenic region of the cerebellum, the DMZ, was studied at the rostral (Figure 7A,B, Appendix A) and caudal (Figure 7C,D) levels. In the rostral region of the cerebellum, GFAP expression was detected in multidirectional fibers radiating from the center of the DMZ (Figure 7A). When labeled with monoclonal antibodies, GFAP+ fibers were clearly outlined in all directions, demonstrating highly specificity of immunolabeling (Figure 7A). Thick basal fibers and thin, sinuous radial fibers were distinguished (Figure 7A). The density of GFAP+ fiber distribution in the rostral region was 115 and 144, respectively. As part of the DMZ, an immunonegative population of NECs was identified in the ML (Figure 7B). In the central part of the DMZ, individual intensely labeled GFAP+ cells, measuring 4–6 µm, were identified in the GrL. The morphometric parameters of GFAP+ cells are presented in Table 3. The quantitative ratio of GFAP+ cells at the rostral and caudal levels is shown in Figure 7Q.

At the caudal level of the cerebellum, the size of the DMZ increased (Figure 7C). The distribution pattern of GFAP+ fibers changed. Thick fibers prevailed in the GrL, where they became noticeably thinner after branching (Figure 7D). Thin GFAP+ fibers often formed organized bundles in the ML (Figure 7C, green triangular arrows), with a higher density in the central part of the DMZ. The immunonegative zone containing NECs was permeated by GFAP+ fibers (Figure 7C). Laterally from the DMZ, the distribution patterns of thick GFAP+ fibers increasingly resembled the branches of climbing cf afferents (Figure 7C). Abundant, reticulated branching of GFAP+ fibers was observed at the border of the ML and GrL (Figure 7D).

Separate GFAP+ aNSPCs clusters were identified in the lateral zone of the cerebellum, in the subpial region of the ML (Figure 7E, Appendix A), as well as individual GFAP+ aNSPCs in the parenchyma of the ML (Figure 7F, Appendix A). The parameters of GFAP+ cells and the intensity of GFAP immunolabeling are presented in Table 3; the quantitative ratio of GFAP+ cells is shown in Figure 7Q.

Cf in juvenile chum salmon also exhibited GFAP immunolabeling (Figure 7E, orange rectangle, G, Appendix A). Cellular targets were not observed at the sites of GFAP+ cf localization, making it impossible to determine which structures they contacted. The endings of GFAP+ cf were mainly located in the GL and in the lowest part of the ML, suggesting that they may contact the somas and dendritic trunks of PCs. Cajal [29] distinguished three phases in the development of cf: the first phase, in which terminal branches are perisomatic; the second, in which these structures contact the upper soma and the proximal dendritic stem; and the third phase, in which they reach their final position on the main dendritic branches of PCs. Similar observations were made in early juvenile trout [26]. The climbing fibers observed in juvenile chum salmon represent the first and second phases of cf development according to Cajal [29] and, thus, may still be relatively immature.

In the basal part of the cerebellum, individual GFAP+ cells and GFAP+ endings of cf were detected in the ML and GrL (Figure 7H). The morphological characteristics of the cells in this zone are shown in Table 3, and the quantitative ratio is shown in Figure 7. Separate GFAP+ cf collaterals were found in the GrL (Figure 7I), as well as terminal extensions of mf, and the central parts of the mossy rosette (Figure 7I, red inset). Studies on early trout juveniles have shown that surface roughness correlates with the degree of maturity of the rosette [26]. The central parts of the rosette, which had reached a certain degree of maturity in juvenile chum salmon, begin to form thin filamentous appendages (Figure 7I, red inset). The mossy-like fibers in adult trout [26], as well as in juvenile chum salmon, exhibit the same characteristics as those observed in mammals. Similar patterns of GFAP+ mf distribution were observed in the mediobasal area of the cerebellum (Figure 7J). Large GFAP+ rosettes of mossy fibers (Figure 7K, blue rectangles) were identified as part of the mediobasal ascending tract, which originates from the lower olivary complex. The GFAP+ cf bundle was located at the base of the cerebellar body, next to the IV ventricle (Figure 7L, Appendix A). In the basolateral regions of the cerebellar body of juvenile chum salmon, large GFAP+ mf rosettes with numerous filamentous appendages were also identified (Figure 7M).

In the GrEm of GFAP+ mf, large and progressively developing rosettes with an irregular surface were formed, representing the central parts of a mossy rosette (Figure 7N, red rectangle). The uneven surface and wide branching indicate a high degree of rosette maturity in the GrEm of juvenile chum salmon. The central part gives rise to numerous thin filamentous appendages, resulting in broad arborization (Figure 7O). The myeloarchitectonic structure of the GFAP+ mf socket reveals a thin, highly structured fiber complex constituting the socket (Figure 7P) within the contact zones of cerebellar glomeruli, where they form synaptic contacts with GrL dendrites, which can also receive terminations from Golgi cell axons. The results of the quantitative analysis of the distribution of GFAP+ cells in various regions of the cerebellum are presented in Figure 7Q. The quantitative study revealed that aNSPCs content in the caudal part of the DMZ was higher than in the rostral part; however, this difference was not statistically significant. Comparative analysis of precursor content in the BMZ and the rostral and caudal DMZ demonstrated significant intergroup differences (*p* < 0.05) (Figure 7Q). Significant differences were also found between the GrL and the rDMZ (*p* < 0.05), as well as between the GrL and the cDMZ (*p* < 0.05), (Figure 7Q). The relative abundance of thin and thick GFAP+ fibers in different cerebellar regions is shown in Figure 7R. The highest number of thin fibers was detected in the GrL (Figure 7R).

Thus, a complex of GFAP+ cells and fibers representing a radial glial complex was identified in the DMZ of the cerebellum of juvenile *O. keta*. The GFAP+ complex of cerebellar afferents shows dorsoventral maturation gradients. In the basal and basolateral parts of the cerebellum, GFAP+ afferents reach a high degree of differentiation, in parallel with the development of the inferior olivary complex. In the GrL, lateral and dorsolateral regions, the degree of maturity of GFAP+ afferent systems is significantly lower than in the basal region. Thus, the DMZ, as the largest neurogenic center of the cerebellum, retains embryonic structural features compared to the basal regions.

### 2.5. Localization of Nestin in the Cerebellum of Juvenile Chum Salmon

Analysis of nestin-immunolabeled sections of the cerebellum of juvenile chum salmon makes it possible to identify small adult-type non-glial progenitor cells (aNSPCs) of rounded shape (4.4 ± 0.5/3.7 ± 0.4 µm) or oval shape (6.8 ± 0.6/5.2 ± 0.6 µm) (Figure 8A–J,M–P). Another type of Nes+ element consisted of granules of various sizes and shapes, not exceeding 3.3 µm, which were detected both inside cells and in the extracellular space (Figure 8A–C). The morphological parameters of nestin-positive and nestin-negative elements are summarized in Table 4.

Individual small, intensely labeled non-glial aNSPCs were identified in the DMZ (Figure 8A), along with both intensely and moderately labeled granules (Figure 8A,B, Appendix A). In the dorsal DMZ and subpial dorsal regions, larger oval, intensely labeled non-glial precursors were identified (Figure 8A,B, Appendix A), along with numerous small rounded nestin+ cells and granules (Figure 8A,C). In the subpial dorsal zone, patterns of massive tangential migration of nestin-negative cells were revealed, accompanied by numerous moderately and intensely labeled nestin granules (Figure 8C, black arrow, Appendix A). A similar distribution pattern of nestin+ cells and granules was observed in the dorsal regions adjacent to the DMZ (Figure 8C, Appendix A) and in more lateral zones (Figure 8D). However, in the dorsolateral regions, radial migration patterns of nestin-negative cells were more typical (Figure 8D, white arrows). The quantitative distribution of nestin+ cells in the dorsal cerebellar zone of juvenile chum salmon is shown in Figure 8Q.

In the lateral cerebellar region of juvenile chum salmon, the distribution patterns of nestin+ cells differed (Figure 8E). The predominant cell type consisted of oval, intensely and moderately labeled non-glial precursors (Figure 8F). Small cells were less common (Figure 8E,G, Appendix A), and nestin+ granules predominated in the subpial region (Figure 8E,G). In some cases, small thickenings containing clusters of nestin+ granules were observed in the subpial zones of the lateral region (Figure 8G, Appendix A). In the deeper layers of the ML of the lateral region, patterns of radial migration of nestin-negative cells predominated, while the number of nestin+ cells decreased (Figure 8H). The quantitative distribution of nestin+ cells in the lateral region is shown in Figure 8Q.

A large population of nestin+ cells forming small clusters was detected in the dorsomedial part of the GrL (Figure 8I, red dotted rectangles). A heterogeneous population of oval and round nestin+ non-glial precursors (Figure 8I), along with evidence of extracellular nestin and granule localization (Figure 8J, green arrows, Appendix A), was found in the area adjacent to the DMZ. In addition to intracellular localization in type 1 and type 2 cells, which sometimes formed clusters of 2–3 cells, nestin immunolocalization was also detected in the extracellular space of the GrL (Figure 8H). The morphometric characteristics and sizes of nestin+ cells are presented in Table 4. The quantitative ratio of nestin+ cell types is shown in Figure 8Q.

Small clusters of type 2 nestin+ cells with strong immunolabeling were observed in the GrEm region (Figure 8K). Individual nestin+ cells and granules were widely distributed throughout the GrL of the GrEm (Figure 8K,L). Single nestin+ cells of types 1 and 2 were also identified in the subpial regions (Figure 8K,L).

In the paramedian basal zone, type 1 and type 2 nestin+ cells forming small clusters were found in the PVZ bordering the IV ventricle (Figure 8M). Separate paired clusters of nestin+ cells were also localized in the SVZ (Figure 8M). Isolated nestin+ type 1 cells were present in the deeper ML regions (Figure 8M). In the lateral basal zone, the PVZ contained multiple cell layers with single and paired type 2 nestin+ cells (Figure 8N). Small clusters of type 2 nestin+ cells were found in the SVZ, while larger clusters and numerous nestin+ granules were detected in deeper parenchymal layers (Figure 8N). A notable feature of this area was the presence of nestin-negative cell clusters in the BMZ and patterns of tangential migration of nestin-negative cells toward the BMZ (Figure 8N,O, Appendix A). In the most lateral areas of the BZ, among the nestin-immunonegative PCs, individual moderately labeled type 1 cells, numerous single nestin+ type 2 cells in ML, and small clusters of granules and type 2 cells in the PVZ were identified (Figure 8P).

The IHC analysis demonstrated broad expression of nestin, an embryonic marker of aNSPCs, in the cerebellum of juvenile chum salmon. Since NEC-like cells expressing nestin were not detected, but widespread nestin expression was found in oval and round cells, localized in the primary and secondary matrix centers (ML, GrL, GrEm), we conclude that the nestin-expressing population in juvenile chum salmon consists of adult-type non-glial precursors producing neurons during homeostatic brain growth, which corresponding intermediate progenitor cells (IPCs) of mammals.

Quantitative analysis showed that type 1 nestin+ cells predominated in the dorsal and dorsolateral zones compared with the basal and lateral regions, although no significant differences were observed (Figure 8Q). The comparative distribution of nestin+ cells in the GrL and GrEm showed significant differences compared to those in the basal and lateral regions (*p* < 0.05) (Figure 8Q).

### 2.6. Vimentin-Producing Cells in the Cerebellum of Juvenile Chum Salmon

Vimentin expression in the dorsal cerebellum was observed in type 1 and type 2 cells, corresponding to adult-type glial precursors (Figure 9A). In the DMZ, vimentin+ cells did not form clusters but appeared as single, intensely labeled oval-shaped aNSPCs (Figure 9A) and small rounded aNSPCs (Figure 9B, Appendix A). Separate large vimentin locations were identified on the border of ML and GL (Figure 9A). A cluster and individual vimentin+ aNSPCs of types 1 and 2 were detected in the dorsolateral area of the ML, including in the surface layers (Figure 9C,D, Appendix A).

Along with aNSPCs, separate subcellular vimentin+ granules were found in the dorsolateral and lateral zones (Figure 9E,F). Morphological parameters of vimentin+ cells and granules are presented in Table 5.

Small accumulations of vimentin+ aNSPCs of both types were detected in the dorsal GrL (Figure 9G,H). Clusters of the type 1 were relatively sparse and composed of type 1 and type 2 aNSPCs, separated by intercellular matrix expressing vimentin (Figure 9G). Clusters of the type 2 formed more compact and dense aggregates, consisting of intensely labeled type 1 and/or type 2 aNSPCs (Figure 9H). An accumulation of type 1 vimentin+ aNSPCs was detected in the GrL adjacent to the DMZ (Figure 9H), corresponding to the localization of type III precursors in ultrastructural sections.

Heterogeneous accumulations of vimentin+ aNSPCs, immunopositive neuropil, and vimentin deposits in the extracellular matrix were detected within the GrL (Figure 9I, Appendix A). Similar extended structures were also found at the border of the GrL and ML (Figure 9J), as well as in the ventrolateral and ventromedial parts of the GrL (Figure 9K,L). These vimentin-producing fields varied in size and length, but generally correspond to neurogenic niches containing adult-type glial precursors.

In the BZ, the distribution of vimentin+ aNSPCs resembled that in the dorsal and lateral zones: intensely labeled type 2 aNSPCs predominated in the ventral part (Figure 9M), while in the medial BZ, type 1 aNSPCs were found in the superficial ML, sometimes forming small clusters (Figure 9N, Appendix A). In general, individual type 1 and type 2 vimentin+ aNSPCs were localized in the ML. Large local neurogenic niches were identified in the dorsal GrEm (Figure 9O), but the number of type 1 and type 2 vimentin+ aNSPCs was reduced compared to the GrL (Figure 9P, Appendix A). Quantitative analysis showed that the highest number of vimentin+ cells was detected in the neurogenic niches of the ventromedial zone, GrL, and GrEm (Figure 9Q), in contrast to early-stage juvenile chum salmon, in which vimentin+ aNSPCs predominantly localized in the dorsal cerebellum [15].

### 2.7. HuCD+ Cells in the Cerebellum of Juvenile Chum Salmon

An extensive population of HuCD+ cells in various anatomical zones was detected in the cerebellum of juvenile chum salmon (Figure 10A–O). Many of these cells were newly formed neurons in the adult cerebellum. Interestingly, a substantial population of HuCD+ cells was found in the dorsal cerebellum of juvenile chum salmon (Figure 10A). Some neurons at the ML–GrL border had developed processes with varicose microcytosculpture, while others within the ML showed no signs of neuronal differentiation (Figure 10B, Appendix A). Neurons in the dorsolateral ML were represented by intensely labeled, mature HuCD+ forms in the upper third of the ML (Figure 10C, Appendix A). On the ML surface, weakly immunolabeled HuCD+ undifferentiated cells and/or cells at early stages of neuronal differentiation, as well as radially migrating weakly labeled HuCD+ cells, were observed (Figure 10C). We propose that homeostatic neurogenesis in the juvenile chum salmon cerebellum is associated with the onset of HuCD expression in cells that have completed proliferation and/or are migrating. It is likely that many of these cells undergo apoptosis before reaching maturity, even during primary bond formation, which aligns with data from other fish species [5,38]. Within the DMZ, the number of HuCD+ cells was relatively low (Figure 10D). Immunopositive cells were detected some distance from the DMZ; in these neurons, HuCD expression levels were moderate or high, but they lacked signs of neuronal differentiation (Figure 10D). The morphometric parameters of HuCD+ cells in the dorsal cerebellar region are presented in Table 6.

An extensive population of HuCD+ cells was detected in the dorsolateral ML (Figure 10E). Small undifferentiated HuCD+ cells (Figure 10E, red arrows) and larger oval HuCD+ single cells, or small clusters, were found in the superficial ML (Figure 10E, yellow arrows). Small undifferentiated cells expressing HuCD obviously correspond to neuroblasts formed as a result of the division of adult-type neuronal precursors. In the upper third of the ML, both types were present, along with small, moderately labeled HuCD+ bipolar cells (Figure 10E, blue arrows). In the deeper layers of the ML, HuCD+ neurons forming connections were identified (Figure 10E, blue dotted rectangle). Patterns of axonal afferent connections labeled with HuCD were present at the border of the ML and GL (Figure 10E, white arrow). Overall, morphogenetic patterns of HuCD expression predominated in the dorsolateral region in various cell types (Figure 10F, Appendix A), indicating a transient stage of primary neuronal differentiation of adult-type precursors. Extensive HuCD labeling in cells of the superficial layers of the dorsolateral cerebellar zone in juvenile chum salmon (Figure 10G) corresponds with data from zebrafish (*D. rerio*) and *A. leptorhynchus* [2], where at least some young HuCD+ cells develop into granular layer neurons [5,38].

The quantitative ratio of HuCD+ cell types in the lateral cerebellum of juvenile chum salmon is shown in Figure 10Q. One-way analysis of variance revealed the presence of intra-group differences (Figure 10Q). Notably, in the dorsal GL, some large HuCD+ cells exhibited axonal arborization (Figure 10H, white arrow, Appendix A), indicating the neuronal differentiation of projection cells. The lateral zone was characterized by a heterogeneous population of HuCD+ cells, in which, in some cases immunolabeled cells predominated in the surface and middle layers of the ML (Figure 10I,J, Appendix A). In other cases, more complex HuCD+ distribution patterns were observed (Figure 10K,L), characterized by accumulations of immunopositive cells in the deep ML layers, forming foci of neuronal differentiation.

In the GrEm, small, intensely labeled HuCD+ cells and their clusters were observed in the superficial GrL regions (Figure 10M,N). Clusters of small HuCD+ cells were localized in the superficial GrEm layers (Figure 10N, Appendix A).

In the basal cerebellum, intense HuCD expression was detected in large projecting bipolar EDCs (Figure 10O,P, Appendix A), forming a heterogeneous HuCD+ cell population in the GL. The morphological characteristics of HuCD+ cells in the BZ are presented in Table 5. The quantitative ratio of HuCD+ cells in the ML, GrL, and GL is shown in Figure 10Q.

## 3. Discussion

Ultrastructural and immunohistochemical studies of the cerebellum in juvenile chum salmon enabled the characterization of several cell populations involved in adult homeostatic neurogenesis. The population of proliferating cells expressing PCNA in various regions of the cerebellar body and eminences was described, while the gliogenic cell population expressing GFAP and vimentin and the neurogenic population expressing nestin were also evaluated. The results of the study allow us to conclude that the TEM-identified cells of types III and IV according to the Lindsey classification [32] represent a proliferating population of intermediate precursors forming clusters in the DMZ and GrL (Figure 3A–C and Figure 6H). Such cells express vimentin (Figure 9B) and probably correspond to interstitial precursors (B cells) of mammals. According to Lindsey’s classification, there is no complete clarity regarding type 1 and type 2 cells; however, it can be assumed that dark type 1 cells in the ML are vimentin+ or GFAP+ and are most likely silent (PCNA-negative), whereas larger type 2 GFAP+ cells represent a proliferating population of progenitors forming clusters in the ML (Figure 7F, Appendix A).

Moreover, a population of neurons expressing HuCD was identified and characterized. IHC labeling data were compared with the TEM and SEM results, allowing for a more detailed ultrastructural characterization of the cell population involved in adult neurogenesis. Ultrastructural analysis of cells and fibers in the ML, GS, and GrL revealed characteristic features of their organization, allowing for the detailed identification and characterization of a heterogeneous population of aNSPCs by comparing ultrastructural traits with IHC results. Together, these data contribute to a more comprehensive identification of aNSPCs phenotypes in the cerebellum of juvenile chum salmon, since the organization of neurogenic niches in fish brains (particularly in the cerebellum) shows significant taxon-specific differences [9,39].

### 3.1. Features of Proliferation in the Cerebellum of Juvenile Chum Salmon

IHC studies showed that an extensive population of proliferating (PCNA+) cells is present in the cerebellum of juvenile chum salmon. The most pronounced proliferative activity was detected in the dorsal and dorsolateral regions of the cerebellum (Figure 6B–D). Quantitative analysis of PCNA+ cells also indicated that the highest proliferative activity occurs in the dorsal zone, as well as in the GrL and GrEm (Figure 6Q).

During IHC analysis of proliferating cells, it was found that cells with the NEC phenotype, as well as glial and non-glial aNSPCs, are PCNA-immunopositive (Table 2). Analysis of IHC and morphometric data showed that among the six identified populations of PCNA+ cells, five types strongly express PCNA. These types correspond to NECs, as well as glial and non-glial precursors of the adult type. Thus, in the process of homeostatic growth, the increase in size of the cerebellum of juvenile chum salmon is carried out both by the embryonic NEC population and by adult populations of aNSPCs, which obviously analogues of the interstitial precursors of the mammalian brain.

Ultrastructural examination revealed that in the dorsal cerebellum both adult glial precursors expressing GFAP and vimentin (Figure 2A–F, Figure 6E,F and Figure 9A–F) and adult non-glial precursors expressing nestin (Figure 3E,F and Figure 9A–H) are present.

Recent comparative studies have demonstrated that cartilaginous fishes *Raja asterias* and *Torpedo ocellata* exhibit differences in both the number of neurogenic niches in the cerebellum and the presence of proliferating cell phenotypes within these niches [9]. Comparative analysis of juvenile *O. masou* also revealed high proliferative activity in the DZ of intact animals, along with a significant increase in proliferation and elevated expression of vimentin and nestin following traumatic brain injury (TBI) [15].

In zebrafish, two distinct populations of proliferating cells in the cerebellum have been identified: the first localized along the dorsal midline, encompassing the vestibular regions, cerebellar body, and the *valvula cerebelli*; the second situated caudo-laterally in the caudal lobe of the cerebellum [40].

Ultrastructural and immunohistochemical analyses have revealed the presence of both adult glial and non-glial precursor cells, differentiated protoplasmic astrocytes, and numerous HuCD+ cells representing differentiated neurons and undifferentiated neuroblasts within the DZ of juvenile chum salmon. Adult-type precursors demonstrate the capacity for proliferation under homeostatic growth conditions and are subject to modulation by various epigenetic factors [39].

Populations of protoplasmic astrocytes, as well as dark cells detected in the cerebellum of juvenile chum salmon, represent various stages of gliogenesis, under certain conditions are also capable of dedifferentiation and proliferation (astrocytes, oligodendrocytes), or transition to proliferative state as a result of the activation of proliferative programs [10,40]. Such stimuli clearly act as triggers that activate dormant aNSPCs or modulate the constitutive cell cycle dynamics characteristic of neurogenic niches [39]. Identifying diverse stimuli that regulate homeostatic proliferation within adult neurogenic niches, alongside their molecular regulators, advances our understanding of the functional roles and biological significance of aNSPCs activity.

### 3.2. Organization of GFAP-Positive Fiber Systems in the Cerebellum of Juvenile Chum Salmon

The determination of GFAP and vimentin-producing systems in the cerebellum of juvenile chum salmon was of particular interest. IHC data for GFAP revealed that the majority of GFAP-labeled elements are fibers of varying diameters, including thin and thick fibers radiating from the DMZ (Figure 7B–D). Small GFAP+ cells were predominantly detected in the dorsomedial zone (Figure 7E,F). According to a previous hypothesis, during migration in the cerebellum of *A. leptorhynchus*, young cells are guided by radial glial fibers delineating migration paths in all three directions [41]. The population of these fibers consists of two subpopulations, one expressing GFAP and the other expressing vimentin. The morphology and distribution of these two populations are similar but not identical, indicating their partial overlap. It is possible that during development, as in the mammalian brain [42], radial glia in juvenile chum salmon initially express vimentin, followed by a gradual decrease in vimentin and a concomitant increase in GFAP expression.

During the development of the nervous system and in the adult central nervous system of mammals and birds, cells with astroglial characteristics (radial glia) function both as neuronal precursors and as scaffolds for cell migration [43,44]. Two glial populations have been identified in the adult zebrafish cerebellum. One consists of cells exhibiting typical radial glial morphology, which are non-proliferative and immunopositive for GFAP, BLBP, and vimentin, localized in the medial zone of the stem cell niche [45]. The processes of these radial glia function as scaffolds for migrating cells. Migrating cells have been observed near blood vessels in the zebrafish cerebellum, apparently using them as scaffolds, consistent with observations in the tegmentum of juvenile chum salmon [17]. Another glial population in zebrafish, immunopositive for S100 but weakly labeled for vimentin, is localized between the ML and GrL [40]. In juvenile chum salmon, GFAP+ glial cells were detected in the similar region between the ML and GrL (Figure 7E). However, in zebrafish, this population expressed gfap: green fluorescent protein (GFP) but did not stain positively with GFAP antibodies. Another interesting feature of juvenile chum salmon is the detection of clusters of small GFAP+ cells in the ML, which indicates their proliferative status. The presence of clusters of type III and IV cells in the GrL, which are obviously vimentin-producing, also suggests that these cell types are capable of proliferation, if not at the time of material fixation, then relatively recently. These results show that in juvenile chum salmon, proliferating cerebellar cells express some canonical markers of RG or astrocytes, in particular, GFAP and vimentin. The results of studies on zebrafish indicate that the proliferating cell population in the cerebellum does not express astrocytic markers such as BLBP, S100, GFAP, and vimentin. Nevertheless, both glial cell populations are associated with the DMZ niche of cerebellar stem cells.

A distinctive feature of the cerebellum in juvenile chum salmon was the detection of GFAP immunolocalization in afferent fiber systems, namely climbing and mossy fibers (Figure 7E). The mf structure in the chum’s cerebellum was characterized by the formation of terminal simple extensions (Figure 7G), detected at different depths of the ML in the dorsolateral region. Studies on early trout juveniles have demonstrated that the surface roughness of rosettes correlates with their degree of maturity [26]. In juvenile chum salmon, the central regions of rosettes that have reached a certain degree of maturity begin to develop thin filamentous appendages (Figure 7I, red inset). Mossy fibers in adult trout [26], as well as in juvenile chum salmon, exhibit features similar to those observed in mammals. Comparable patterns of GFAP+ mossy fiber distribution were identified in the mediobasal cerebellar area (Figure 7J). Large GFAP+ mossy fiber rosettes (Figure 7K, blue rectangles) were identified as components of the mediobasal ascending tract, with the lower olivary complex as their apparent source. The GFAP+ climbing fiber bundle was observed at the base of the cerebellar body, adjacent to the fourth ventricle (Figure 7L). Large GFAP+ mossy fiber rosettes, forming numerous filamentous appendages, were detected in the basolateral regions of the cerebellar body of juvenile chum salmon (Figure 7M). Particular attention was given to the detection of highly differentiated mossy fiber rosettes and glomeruli in the GrEm, where GFAP+ cells were absent in juvenile chum salmon (Figure 7N–P).

Climbing fibers in the cerebellum of juvenile chum salmon were also GFAP-immunopositive (Figure 7E, orange rectangle). GFAP immunolocalization patterns of climbing fibers were observed in both the BZ and GrL, as well as in the lower third of the molecular layer (ML), suggesting contact with the somata and dendritic trunks of Purkinje cells. TEM data revealed dense interactions between the dendritic trunks of Purkinje cells and climbing fibers (Figure 2C), corroborating earlier findings in juvenile trout [26] and indicating ongoing development of climbing fibers concurrent with Purkinje cell dendrites.

Ultrastructural and IHC analyses revealed the presence of a GFAP+ radial glial complex within the DMZ of the juvenile chum cerebellum. Simultaneously, the GFAP+ complex of cerebellar afferents develops with a differentiation gradient along the dorsoventral axis. In the basal and basolateral cerebellar regions, GFAP+ afferents exhibit a high degree of differentiation, concomitant with the maturation of the inferior olivary complex. Conversely, the degree of maturity of GFAP+ afferent systems in the GrL, lateral, and dorsolateral regions is labeled lower than in basal areas. Thus, the DMZ, as the cerebellum’s largest neurogenic center, retains embryonic structural characteristics relative to the basal regions.

### 3.3. Ultrastructural and Immunohistochemical Identification of Adult Neural Stem/Progenitor Cells (aNSPCs) in the Cerebellum of Juvenile Chum Salmon

Ultrastructural analysis of the DMZ in juvenile chum salmon revealed individual cells exhibiting a NEC phenotype (Figure 3B), indicating the predominance of adult-type precursors at this ontogenetic stage. Importantly, PCNA immunolabeling strongly indicates a proliferating NEC population within the DMZ (Figure 6F,G). Our findings are consistent with data from zebrafish, in which NECs persist into adulthood [40]. Zebrafish cerebellar stem cells maintain neuroepithelial properties, including ventricular contact, apical-basal polarity, and distinct adhesive junctions. During ultrastructural analysis of the DMZ of juvenile chum, we identified cells with a similar morphology (Figure 3A). In mammals, neural precursors exhibit neuroepithelial properties, specifically expressing nestin and retaining apical-basal polarity, including adhesive junctions and localization of centrosomes [45].

SEM revealed numerous superficially located rounded cells on the dorsal and dorsolateral cerebellar surfaces of juvenile chum salmon, representing both resident quiescent (Figure 5A,E,F) and proliferating (Figure 5B–D) populations. Previous SEM studies of the fish cerebellum primarily focused on microcytosculpture characteristics and cellular composition [46].

Ultrastructural analysis of ML cells of juvenile chum showed the presence of three cell types (types I, II and III), which are glial aNSPCs of the adult type (Figure 2C–F). Comparison of TEM, SEM, and IHC vimentin labeling data suggests that individual vimentin+ cells localized in the ML may correspond to type III cells, as well as to dark cells (type I) representing aNSPCs. However, it should not be excluded that the replacement of the vimentin-expressing system in the cerebellum with a GFAP-expressing one also presupposes the emergence and development of GFAP+ resident small undifferentiated dark cells, as well as larger type II GFAP+ cells capable of proliferation. Such GFAP+ elements identified by us in the ML of juvenile chum are a definite proof of ontogenetic transformation.

In the granular layer (GrL), clusters of glial precursors of types III and IV were identified (Figure 4B–E), forming groups of varying composition and localizing in distinct GrL regions. TEM findings correlate with vimentin immunolabeling data within the granular layer (Figure 9G–L). Diffuse accumulations of vimentin+ cells detected in the central, lateral, and basal granular zones correspond to TEM-identified clusters of types III and IV precursors (Figure 4C–F). Consequently, neurogenic niches containing adult-type glial precursors are present in these GrL regions of juvenile chum salmon. PCNA immunolocalization patterns suggest that while some niches harbor proliferating cells, the majority predominantly contain quiescent precursors.

Ultrastructural examination of adult non-glial progenitors identified in the DMZ (Figure 3A,E,F) and the dorsal granular layer (GrL) (Figure 4A) was compared with the distribution of nestin+ cell clusters observed in the corresponding locations (Figure 8I,J). Nestin, an intermediate filament protein, is a widely used neural stem cell (NSC) marker and is robustly expressed by mammalian cerebellar precursors [47]. In the juvenile chum salmon cerebellum, nestin expression is observed in proliferating and migrating cells, consistent with adult zebrafish data, where nestin is also present in S100+ glial cells [40].

Ultrastructural and immunohistochemical analyses of juvenile chum revealed that specific GrL regions containing non-glial precursors likely correspond to nestin+ cell clusters, indicative of neurogenic niches generating neuronal precursors. The presence of individual nestin+ cells within the molecular layer across various cerebellar topographic zones is associated with quiescent or proliferating neuronal precursors under homeostatic conditions, consistent with observations in adult zebrafish [40].

Comparative analysis of nestin+ and HuCD+ cell populations in the dorsal and dorsomedial cerebellar regions of juvenile chum salmon suggests that nestin+ non-glial precursors give rise to numerous HuCD+ neuroblasts prior to the establishment of intercellular connections. These cells correspond to a population of interstitial intermediate progenitors (B cells) in mammals. The abundant presence of PCNA+ cells in the dorsolateral and lateral cerebellar regions corresponds with elevated nestin+ cell production, indicating a substantial proliferative activity of neuronal precursors, some of which differentiate into HuCD+ neuroblasts at the initial stages of neuronal differentiation. Nestin expression in neurogenic niche cells has also been documented in zebrafish [40]. Moreover, it is important to consider that adult-type progenitors, both glial and neuronal, exhibit considerable migratory capacity, as demonstrated by tangential and radial migration patterns observed in the molecular layer.

Collectively, immunohistochemical and ultrastructural evidence suggests that proliferation of adult-type glial and non-glial precursors expressing vimentin and/or GFAP, and nestin and/or HuCD, occurs within the DMZ, ML, and GrL of the juvenile chum salmon cerebellum under homeostatic conditions. Neurogenic activity is mediated by individual glial and non-glial aNSPCs within the ML, as well as by neurogenic niches in the GrL, including the dorsal granular niche and DMZ harboring non-glial aNSPCs, alongside lateral, medial, and basal niches containing glial aNSPCs.

Quantitative analysis showing the ratios of PCNA+, vimentin+, and nestin+ cells are presented in Figure 11A. These ratios reflect the potential proliferative activity of glial and non-glial aNSPCs under homeostatic growth conditions. Comparative ratios of HuCD+, nestin+, and PCNA+ cells (Figure 11B) reveal correlations among proliferating cells, neuroblasts, and non-glial progenitors. Thus, features of the ultrastructural organization of NECs localized in the DMZ and representing a limited population of active contributors of adult neurogenesis have been established in the cerebellum of juvenile chum salmon. The predominant active contributors to neurogenesis in the juvenile chum salmon cerebellum include populations of adult-type glial precursors expressing vimentin and GFAP, as well as adult-type non-glial precursors expressing nestin.

### 3.4. Ultrastructural and Immunohistochemical Identification of Efferent Deep Cells (EDCs) in the Juvenile Chum Salmon Cerebellum

Ultrastructural analysis revealed a population of cells in the GL that lacked dendritic bouquets in the ML (Figure 2A,B). These cells were observed in contact with protoplasmic astrocytes or type II glial precursors (Figure 2E). IHC labeling for HuCD enabled identification of large projection neurons in the lateral zone of the granular layer (Figure 10G,H) and basal zone (Figure 10O,P) of the cerebellar body, classified as EDCs. Based on their axonal and dendritic arborization patterns, these cells can be classified as broad-dendritic neurons. Previously, large projection neurons in the GL expressing parvalbumin were detected in the cerebellum of rainbow trout *O. mykiss* [47], Mediterranean barbel *Barbus meridionalis* [48] and juvenile *O. masou* [49]. Studies have shown that the parvalbumin-producing neurons in trout [48,49] and *O. masou* [50] are PCs. Conversely, in tench *Tinca tinca* and other Cyprinidae species, parvalbumin expression is restricted to cells in the vestibulolateral lobes and medial flap region, while cells in the cerebellar body and lateral flap are parvalbumin-immunonegative [51]. Antibodies against parvalbumin and calretinin, combined with Nissl fluorescent staining, have been used to identify Purkinje cells and EDCs in the zebrafish cerebellum [39].

Differences in parvalbumin (PV) immunolocalization between salmonid and cyprinid fish may reflect functional heterogeneity among Purkinje cell populations in fish brains [49,50]. In the cerebellum of juvenile *O. masou*, PV immunolocalization correlates with GABAergic and CBS+ neurons [50]. PCs are not efferent systems in the fish brain, but are associated with EDCs projecting into the tegmentum, octavolateral nuclei, and spinal cord [52]. Despite differences in cytoarchitecture and projection patterns between bony fish and mammalian cerebella, parvalbumin immunoreactivity exhibits evolutionary conservation. In particular, PV reactivity in PCs is present in both fish and mammals, whereas PV-immunonegative EDCs in juvenile *O. masou* [50] are innervated by PV+ PCs axons.

Functionally, EDCs are analogous to mammalian cerebellar nuclei [53], which are also PV-immunonegative, but receive PV+ signals from PCs [54,55]. PCs axons are formed at the initial stages of development of rainbow trout fry, and PCs axon collaterals on EDCs are accumulated 4 days after hatching [26]. Subsequently, calcium-dependent mechanisms are activated, facilitating axonal outgrowth, synapse formation, and neural signal transmission. The results of studies on juvenile chum salmon show the presence of HuCD+ axon collaterals of EDCs, which display extensive arborization in the lower third of the ML. HuCD immunopositivity in juvenile chum salmon EDCs is linked to calcium signaling activation and subsequent projection development. EDC identification per Nieuwenhuys relies on axon collateral branching patterns, ultrastructural cytoplasmic characteristics, and afferent synapse features [53]. The results of ultrastructural studies on juvenile chum salmon allowed us to establish the features of the organization of EDCs in the dorsal and basal regions of the cerebellum. The conducted studies on juvenile chum salmon can become a starting point for studying the biology and projection properties of EDCs identified in the cerebellum using TEM and IHC verification of HuCD.

### 3.5. Limitations

The conducted studies, while providing information on the homeostatic growth of juvenile chum salmon and suggesting ultrastructural and IHC characteristics of various types of cells and fibers in the cerebellum, nevertheless have a number of limitations. One limiting factor is the dependence on correlative markers (PCNA+ for proliferation, HuCD+ for differentiation) to determine cell behavior without accompanying functional verification. Although these indicators provide valuable initial information, they inherently do not allow us to establish cause-and-effect relationships or trace the development of a particular family tree. Another limitation may be the lack of data in the materials of this work on the immunoelectronic labeling of glial and non-glial progenitors, which makes it possible to unambiguously interpret the immunopositive and electron microscopic profiles of cells in determining their final phenotypes.

## 4. Materials and Methods

### 4.1. Experimental Animals

The experiments were conducted on 20 intact juvenile chum salmon (*O. keta*), one year old, with body lengths ranging from 18 to 31 cm and weights from 41 to 49 g. The animals were provided by the Ryazan Experimental Fish Hatchery (Ryazanovka, Russia) in 2023. Most of the fish used in this study were males. The salmon were maintained in aerated freshwater aquaria at 13–14 °C and fed once daily. The light–dark cycle was 14/10 h. The concentration of dissolved oxygen in the water was 7–10 mg/dm^3^, which corresponds to normal saturation. All experimental procedures with animals complied with the regulations established by the statute of the A.V. Zhirmunsky National Scientific Center for Marine Biology of the Far Eastern Branch of the Russian Academy of Sciences (NSCMB FEB RAS) and were approved by the Ethical Commission on the Humane Treatment of Experimental Animals (approval No. 14, dated 21 July 2025, at the Biomedical Ethics Commission meeting of NSCMB FEB RAS). The animals were divided into seven groups.

### 4.2. Transmission Electron Microscopy

The ultrastructural profiles of neurons and glia in the cerebellar body were studied using transmission electron microscopy. For histological control, semithin sections of the cerebellar body, as well as granular eminences in the frontal plane, were examined. Semithin sections with a thickness of 500 μm were mounted on slides at the corresponding rostrocaudal level of the cerebellum. Semithin sections were stained with 1% methylene blue in a 1% aqueous solution of sodium tetraborate.

After fixation of the cerebellum in 2.5% glutaraldehyde solution in 0.2 M cacodylate buffer (pH 7.4) at a temperature of 4 °C overnight, the samples were washed and then post-fixed for 2 h with 1% OsO4 solution in 0.2 M cacodylate buffer. After that, the tissues were washed and dehydrated in an ascending series of ethanol, and then soaked in 100% ethanol and LR-White resin. The next day, the tissues were soaked in fresh LR-White resin twice for a 6-h periods, and then embedded in gelatin capsules and polymerized at 40 °C.

Using a Leica UC7 ultramicrotome (Wetzlar, Germany), semithin sections with a thickness of 500 μm were mounted on slides, and ultrathin sections with a thickness of 50–60 nm were mounted on formvar-coated copper meshes at the corresponding rostrocaudal level of the cerebellum. Semithin sections were stained with 1% methylene blue in a 1% aqueous solution of sodium tetraborate. Ultrathin sections were stained with 2% aqueous solution of uranyl acetate and lead citrate. Visualization was performed using a Zeiss Sigma 300 VP transmission electron microscope (Carl Zeiss, Cambridge, UK) and the AMT Image Capture Engine software (version 5.44.599).

To analyze the morphology of cells, present in the niches of the molecular and granular layers of the cerebellar body, seven ultrastructural profiles (Types IIa, IIb, III, IVa, IVb, V and VI), previously described in the work “Neurogenic niches of the forebrain” [56] were used.

### 4.3. Cell Counting and Visualization

Visualization of semithin sections of the cerebellar body, and granular eminences was performed using an AxioImager Z2 microscope (Carl Zeiss, Göttingen, Germany). To study the ultrastructural organization of the cerebellum, low-power (2–4K) TEM images were acquired at rostrocaudal levels.

These images were used to identify the topographic regions of the cerebellum: the molecular layer, DMZ, ganglion layer, infra- and extraganglionic plexuses, and granular layer, for subsequent cell type analysis. From 5 to 15 cells of each morphologically distinct type were selected for morphological analysis across different layers of the cerebellum (molecular, granular, and ganglion layers) and the DMZ. If only a few cells of a given morphology were observed within the DMZ, all identified cells were analyzed. The following features were studied to characterize cell types: contour, color, chromatin organization, number of nucleoli, length of the long and short axes of the nucleus, percentage, presence of mitochondria, cilia, microvilli, vacuoles, lipid droplets, dense bodies in the cytoplasm, localization of cell types, and intercellular contacts. Using these criteria, the morphological profiles and frequencies of each cell type within the cerebellar layers (ML, GrL, GL), the DMZ, and as percentages of all studied cells were calculated. Based on the identified morphological features, a model of cellular organization of the cerebellum and a classification scheme for cell types comprising different niches were developed.

### 4.4. Scanning Electron Microscopy

Brain samples of chum salmon *O. keta* were fixed whole in 2.5% glutaraldehyde solution prepared in 0.2 M cacodylate buffer (pH 7.4) without NaCl for 24–48 h at 2–4 °C. After that, the samples were washed in 0.1 M cacodylate buffer for 20 min using a Biosan Mini-Shaker PSU-2T (Riga, Latvia). The samples were then dehydrated through a graded ethanol series, followed by progressive substitution with pure acetone [57]. Initially, samples were sequentially immersed for 10 min each in ethanol solutions of 7%, 15%, 30%, 50%, 70%, 80%, 90%, and 96%. Then, the samples were incubated for 10 min in mixtures of ethanol and acetone with ratios of 3:1, 1:1, and 1:3, progressively transitioning to pure acetone. Finally, the samples were dried using a BAL-TEC Critical Point Dryer 030 (Pfeffikon, Switzerland). Dried cerebellum samples were placed on the surface of aluminum tables, pre-coated with carbon tape. Next, the samples were chromium-sprayed using a Q 150T ES vacuum coating device (London, UK) for thin membranes. Next, the structural features of the *O. keta* cerebellum were examined using a Sigma 300 VP Carl Zeiss scanning electron microscope. Various brain parameters were measured using the Smartiff program. All resulting micrographs were processed using the Adobe Photoshop CS6 graphics program, beta version (22 March 2012) for Microsoft Windows.

### 4.5. Preparation of Material for Immunohistochemical Studies

*Anesthesia, prefixation of intact animals*. The fish were anesthetized with 0.1% tricaine methanesulfonate (MS-222) solution (Sigma, St. Louis, MO, USA, Cat. No. WXBC9102V) for 10–15 min. After anesthesia, the intracranial cavity of the immobilized animals was perfused with a 4% solution of paraformaldehyde (PFA, BioChemica, Cambridge, MA, USA; Cat. No. A3813.1000; lot 31000997) in 0.1 M phosphate buffer (Tocris Bioscience, Minneapolis, MN, USA; Catalog number 5564, Lot No. 5, pH 7.4). The animals were euthanized by rapid decapitation. After preliminary fixation, the brain was removed from the cranial cavity and fixed for 24 h in a 4% paraformaldehyde solution in 0.1 M phosphate buffer. The samples were then kept in a 30% sucrose solution at 4 °C for two days (with seven solution changes). Serial frontal (50 μm thick) sections of the chum salmon brain were cut out on a freezing microtome (Cryo-star HM 560 MV, Waldorf, Germany). Every third frontal section of the cerebellum was taken for the reaction, to obtain representative data.

### 4.6. Immunohistochemical Verification of PCNA, GFAP, Vimentin, Nestin and HuCD

The expression of markers PCNA, GFAP, vimentin, nestin, and HuCD in intact juvenile chum salmon *O. keta* was studied. Before IHC, endogenous peroxidase activity and nonspecific staining (background) were blocked by incubation with 1% hydrogen peroxide for 20 min at room temperature. To eliminate nonspecific staining, brain regions were incubated with nonimmune equine serum.

Immunoperoxidase labeling was performed using monoclonal mouse antibodies against PCNA, GFAP, vimentin, nestin, and HuCD at a dilution of 1:300 on frozen free-floating brain slices at a temperature of 4 °C for 48 h (Table 7).

The ABC complex system (Vectastain Elite ABC kit; Vector Laboratories, San Francisco, CA, USA; Catalog No. PK-6100) and a red substrate (VIP Substrate Kit; Vector Labs, Burlingame, CA, USA; Catalog No. SK-4600) were used for visualization. The sections were washed in a phosphate buffer (pH 7.2) three times for 10 min after each incubation stage to remove unbound antibodies and prevent the development of background coloration. The staining was monitored under a light microscope to prevent an amplification of the reaction. The preparations were additionally stained with methyl green (Bioenno, Livescience, Santa Ana, CA, USA; Cat No. 003027) to identify immune-negative cells, which made it possible to differentiate cell populations. After the reaction was completed, the sections were washed in three changes of distilled water to remove residual salts.

Sections of the cerebellum were placed on polylysine-coated slides (BioVitrum, St. Petersburg, Russia) and left to dry completely. To identify immunonegative cells, cerebellar sections were additionally stained with 0.1% methylene blue solution (Bioenno Livescience, Santa Ana, CA, USA, Cat. No. 003027). The color development was observed under a microscope. The sections were washed in three changes of distilled water for 10 s, then differentiated for 1–2 min in a 70% alcohol solution, and finally for 10 s in 96% ethanol. The slices were dehydrated according to the standard procedure: two xylene changes of 15 min each. They were then placed under cover glasses in a Bio-optica mounting medium (Milan, Italy). The method of negative control was used to assess the specificity of the immunohistochemical reaction.

Cerebellar sections, instead of being incubated with primary antibodies, were treated with a 1% solution of nonimmune equine serum for one day and processed in the same way as the sections incubated with primary antibodies. No immunopositive reaction was observed in any of the control experiments.

### 4.7. Microscopy

For visualization and morphological and morphometric analysis of cell body parameters (measuring larger and smaller soma diameters), a research-grade inverted microscope with a contrast enhancement nozzle was used: the Fluorescence Microscope Axiovert 200 M with ApoTome fluorescence module and AxioCam MRM and AxioCam HRC digital cameras (Carl Zeiss, Jena, Germany). The material was analyzed using AxioVision software (version 4.8) with the Axiovert 200 M microscope (Carl Zeiss, Jena, Germany). The measurements were carried out at magnifications of 100×, 200× and 400× and in several randomly selected fields of view for each area of interest. The number of labeled cells in the visual field was calculated at a 200-fold magnification. Micrographs of the preparations were taken with an Axiovert 200 digital camera. The material was processed using the AxioVision software (version 4.8; Zeiss, Jena, Germany) and Corel Photo-Paint 17 graphic editor.

### 4.8. Densitometry

The optical densities (ODs) of IHC-labeled products in neuron bodies and immunopositive granules were measured using an Axiovert 200 M microscope with AxioVision software (version 4.8; Zeiss, Jena, Germany). The Wizard program was used to perform a standard optical density assessment for 5–7 slices by selecting 10–15 intensely/moderately labeled and immunonegative cells of the same type for analysis. Then, the average optical density value for each cell type was subtracted from the maximum optical density value for immunonegative cells (background), and thus the actual value in relative optical density units (UODs) was obtained.

### 4.9. Stereological Method

To obtain reliable quantitative characteristics of the cerebellum, a stereological approach based on a sample of serial slices was used. This method made it possible to estimate the distribution and density of immunopositive cells in various areas of the cerebellum, excluding systematic errors. Areas of interest included the granular layer, the molecular layer, and granular eminences. The data made it possible to reconstruct volumetric structures and evaluate the spatial distribution of cells. The calculations were performed using a systematic random sampling method, which ensured high accuracy of the analysis. The use of stereological methods made it possible to quantify both the density of cells and their distribution over individual areas of the cerebellum.

### 4.10. Statistical Analysis

Quantitative data were processed using Statistica 12 (version 12, StataCorp LP., College Station, TX, USA) and SPSS. (version 12.0; SPSS Inc., Chicago, IL, USA). All results were presented as mean ± standard deviation (M ± SD).

One-way analysis of variance (ANOVA) with Bonferroni correction was used to analyze the differences between groups. Values of *p* < 0.05 were considered statistically significant. Additionally, correlation analyses were performed to identify relationships between morphological changes and experimental conditions.

## Figures and Tables

**Figure 1 ijms-26-09267-f001:**
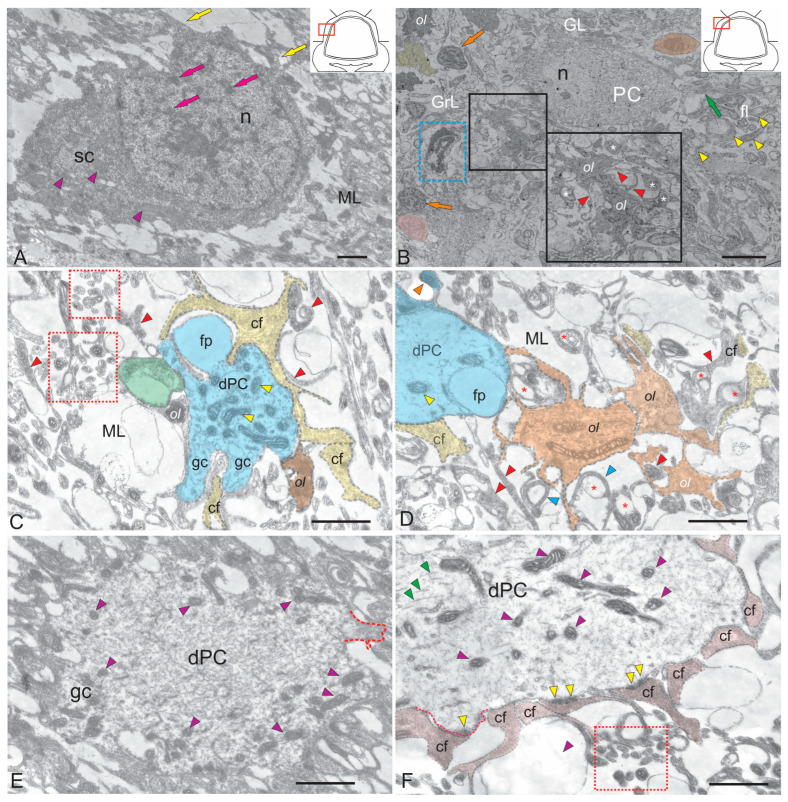
Ultrastructural organization of cells in the dorsal region of the molecular layer of the cerebellar body of juvenile chum salmon *Oncorhynchus keta*. (**A**) Spherical microneuron of the stellate type (SC) with short dendrites (yellow arrows); n—nucleus, pink arrows indicate blocks of heterochromatin, mitochondria are indicated by purple triangular arrows. (**B**) General view of the dorsal region, including the molecular layer (ML), containing parallel fibers (green arrow), fibrous layer fibers (fl, indicated by yellow triangular arrows); pear-shaped PCs are indicated by the arrow in the GL; n—nucleus; oligodendrocytes (ol) are shown in the black rectangle and inset; the processes are indicated by red triangular arrows; a white asterisk indicates forming myelin fibers, the blue dotted rectangle shows the ends of cf; in the granule cells (GrCs), the orange arrows indicate type IV glial precursors; the oligodendrocyte in a state of mitosis is highlighted in yellow; granule cells are indicated by pink arrow. (**C**) Growth cones (gc) and filopodia (fp) on PC dendrites (dPC, shown in blue) climbing fibers (cf) adjacent to dPC indicated by yellow arrows, oligodendrocyte (ol) is shown in orange, mitochondria are indicated by yellow triangular arrows; cf terminal branches are shown in red dotted rectangles; cf endings indicated by red triangular arrows; myelin fiber is shown in green. (**D**) Cluster of oligodendrocytes (shown in orange) forming myelin sheaths (triangular blue arrows) on the cf surface (indicated by red asterisks); filopodia is indicated by orange triangular arrows; mitochondria are indicated by yellow triangular arrows; cf endings are indicated by red triangular arrows. (**E**) The dendritic bouquets of PCs covered with spikes (highlighted by red dotted line), mitochondria are indicated by purple triangular arrows, the remaining designations are as in (**C**). (**F**) Terminations (shown in pink) on the dPC forming asymmetric excitatory type synapses are indicated by yellow triangular arrows; mitochondria—purple triangular arrows, neurofilaments—green triangular arrows, a spike on the surface of the dPC is highlighted with a red dotted line, cf terminal branches are shown in the red dotted rectangle. Scale: (**A**) 2 µm; (**B**) 10 µm; (**C**–**F**) 1 µm.

**Figure 2 ijms-26-09267-f002:**
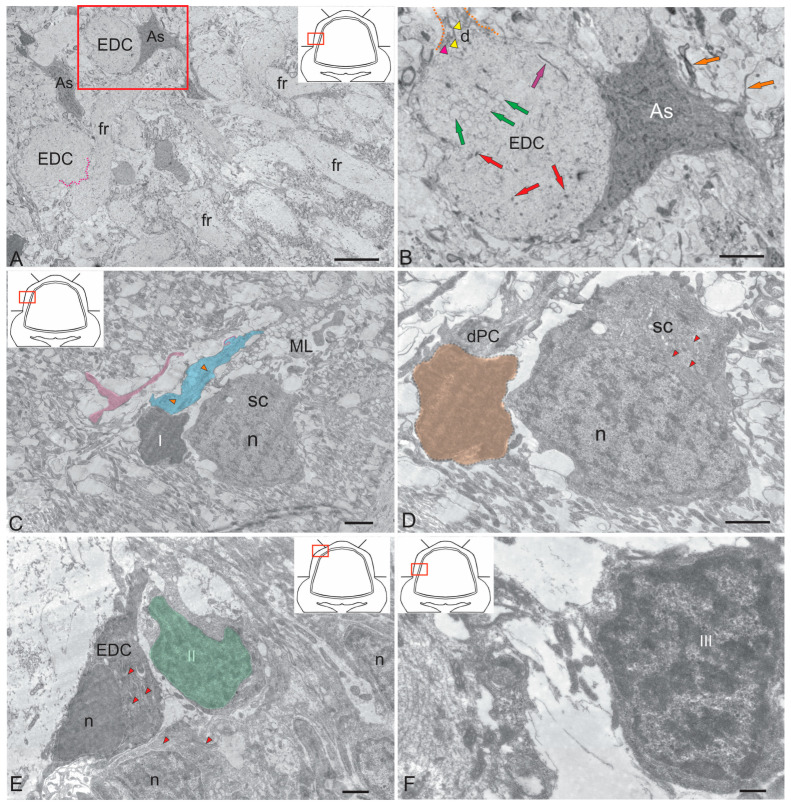
Ultrastructural organization of neurons and glia in the molecular layer of the cerebellar body of juvenile chum salmon *Oncorhynchus keta*. (**A**) An astrocyte (As) of the protoplasmic type in contact with a eurydendroid cell (EDC); dark pink dotted line indicates the perisomatic process of the EDC; fr—fibrous fibers. (**B**) The fragment indicated by the red rectangle in (**A**) at higher magnification; a thin filamentous fibrillation in the cytoplasm of the EDC is indicated by a purple arrow, glycogen granules (red arrows), lipid droplets (green arrows); the origin of the proximal dendrite (d) is outlined by an orange dotted line; microtubules (yellow triangular arrows), subsurface cisterns (dark pink triangular arrow), orange arrows indicate processes of the astrocyte. (**C**) A dark cell (I) in contact with stellate cells (SC) in the molecular layer (ML); a T-shaped fiber is shown in pink; n—nucleus, PC dendrites (dPC) are highlighted in blue; dPC mitochondria are indicated by orange triangular arrows. (**D**) SC in contact with the dark cell I (highlighted in orange) at higher magnification; n—nucleus, mitochondria are indicated by red triangular arrows. (**E**) A type II glial cell (shown in green) located next to the developing EDC; mitochondria are indicated by red triangular arrows, other designations are as in (**D**). (**F**) An adult type glial precursor (aNSPCs, III) according to Lindsay’s classification. TEM. Scale: (**A**) 10 µm; (**B**–**D**) 2 µm; (**E**) 1 µm; (**F**) 400 nm.

**Figure 3 ijms-26-09267-f003:**
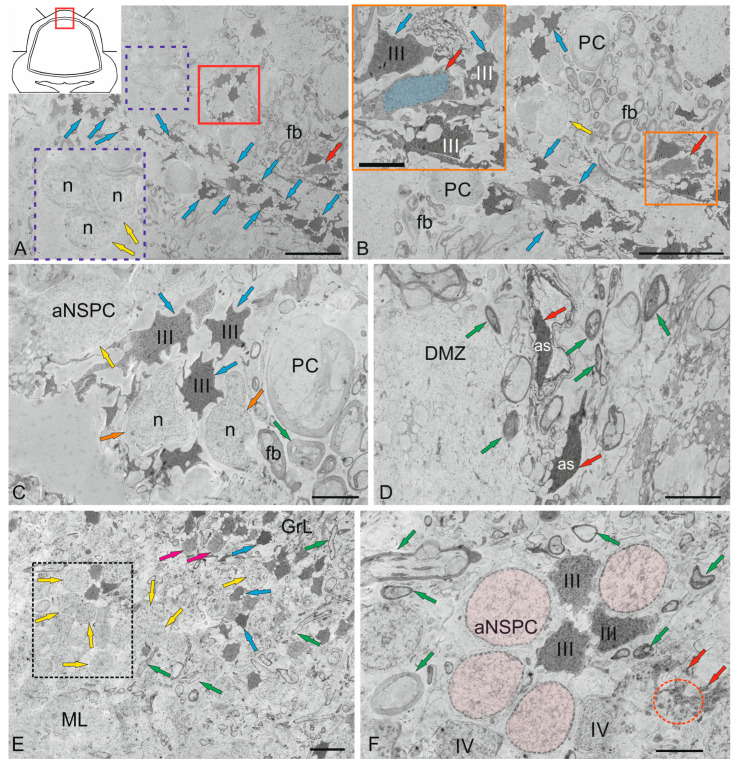
Ultrastructural organization of the dorsal matrix zone of the cerebellum of juvenile chum salmon *Oncorhynchus keta*. (**A**) A general view of the dorsal matrix zone (DMZ), shown in the red square of the pictogram, which includes a cluster of non-glial adult-type neural stem/progenitor cells (aNSPCs, yellow arrows) in the dorsomedial region (blue dotted inset); a ventral cluster of aNSPCs (in the blue dotted square); individual aNSPCs are indicated by blue arrows; patterns of sprouting myelin fibers (fb, red arrows); neuroepithelial cells (NECs, red arrows). (**B**) An enlarged fragment (in orange inset) showing details of the ultrastructure of the NECs (red arrows), surrounded by type III aNSPCs (blue arrows); non-glial adult-type neural stem/progenitor cells (yellow arrows) Purkinje cells (PCs) are indicated. (**C**) An enlarged fragment showing details of the ultrastructure of a heterogeneous cluster in the blue dotted square in (**A**); type III aNSPCs are indicated by blue arrows; non-glial type aNSPCs are indicated by yellow arrows; embryonic-type intermediate precursors are indicated by orange arrows; myelin fibers (fb, green arrows); Purkinje cells (PCs). (**D**) Organization patterns of perivascular astroglia (as), indicated by red arrows; fibers are indicated by green arrows. (**E**) aNSPCs of non-glial type (yellow arrows) form clusters in the ML (in a black dotted rectangle) and at the border of granular eminence (GrEm), type III aNSPCs—blue arrows, type IV aNSPCs crimson arrows, cf (climbing fibers)—green arrows. (**F**) An enlarged fragment in a black dotted rectangle in (**E**), myelin fibers—green arrows; non-glial type aNSPCs are highlighted in pink; type III aNSPCs (III); type IV aNSPCs (IV); secretory granules are indicated by red arrows; a cluster of secretory granules is outlined by a red dotted oval. TEM. Scale: (**A**,**B**) 20 µm; (inset); (**E**) 10 µm; (**C**,**D**,**F**) 5 µm.

**Figure 4 ijms-26-09267-f004:**
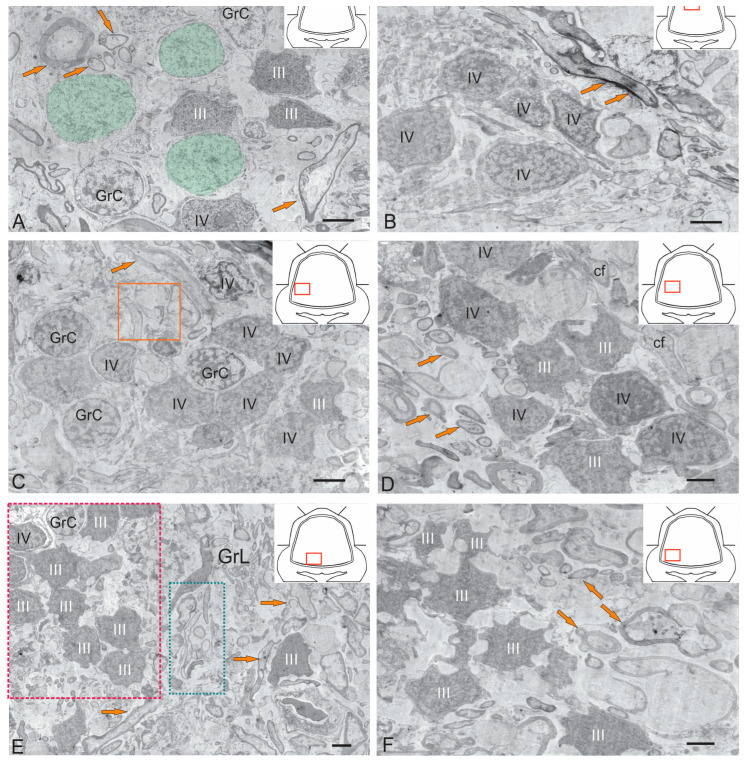
Ultrastructural organization of the granular layer of the cerebellum of juvenile chum salmon *Oncorhynchus keta*. (**A**) Cells of the granular layer (GrL) of the dorsomedial part of the cerebellar body; granule cells (GrCs); type III adult-type neural stem/progenitor cells (aNSPCs)—III, type IV aNSPCs—IV; climbing fibers (cf)—orange arrows; non-glial progenitor cells are highlighted in green. (**B**) In deeper layers of GrL, type IV aNSPCs (IV); unmyelinated fibers are indicated by orange arrows. (**C**) Diffuse distribution patterns of GrCs and aNSPCs of types III and IV in the lateral part of the cerebellar body; myelinated fibers are indicated by orange arrows; mixed fibers are shown in the orange rectangle. (**D**) Cluster distribution patterns of types III and IV aNSPCs in the ventrolateral part of the cerebellar body; climbing fibers (cf); myelinated cf are indicated by orange arrows. (**E**) Constitutive neurogenic niches of mixed adult type (in a dark red dotted rectangle) type III aNSPCs (III) in the medio-basal zone of the cerebellar body; a cluster of climbing fibers (cf, in a green dotted rectangle); myelinated cf are indicated by orange arrows. (**F**) Adult neurogenic niche formed by type III aNSPCs from the ventrolateral part of the cerebellar body; myelin fibers—orange arrows, other designations are as in (**E**). TEM. Scale: (**A**–**C**) 3 µm; (**D**–**F**) 2 µm.

**Figure 5 ijms-26-09267-f005:**
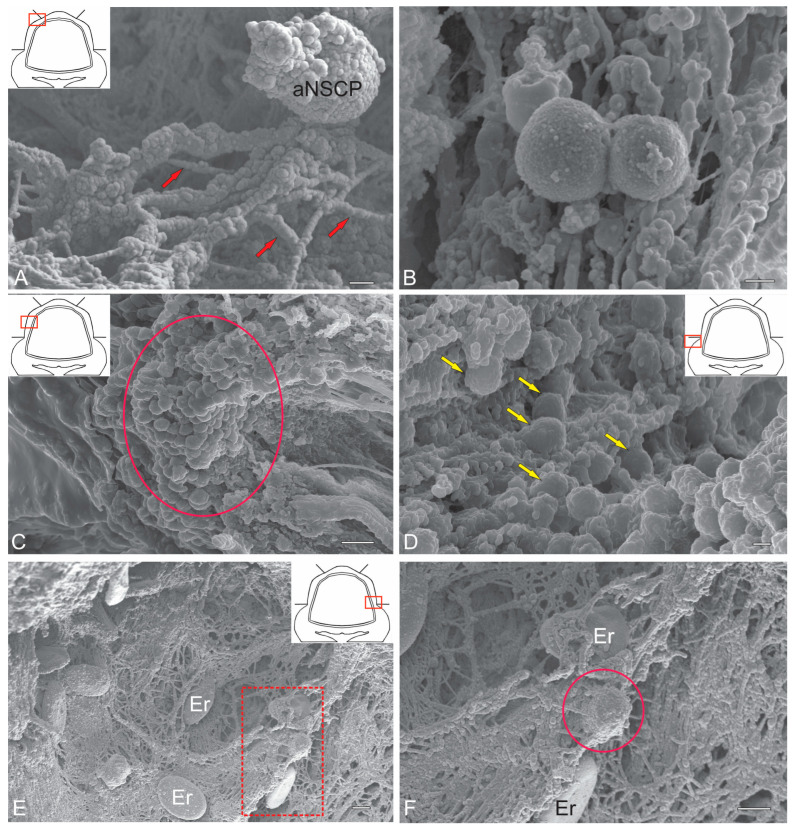
Stereoscopic organization of adult neural stem and progenitor cells (aNSPCs) in the cerebellum of juvenile chum salmon *Oncorhynchus keta*. (**A**) aNSPCs are fixed by microfibrils (red arrows) on the surface of the molecular layer. (**B**) Cluster of paired aNSPCs. (**C**) Large stromal clusters of aNSPCs (in a red oval) in the dorsolateral regions of the cerebellum. (**D**) Diffuse patterns of aNSPCs distribution (yellow arrows) in the granular eminence (GrEm) associated with the surface matrix. (**E**) aNSPCs are fixed by microfibrils (in the red dotted rectangle) on the surface of the molecular layer; an erythrocyte (Er). (**F**) An enlarged fragment from the red dotted rectangle in (**E**); a single cluster of aNSPCs (bounded by a red oval). SEM. Scale: (**A**) 1 µm; (**B**,**D**) 2 µm; (**C**) 10 µm; (**E**,**F**) 3 µm.

**Figure 6 ijms-26-09267-f006:**
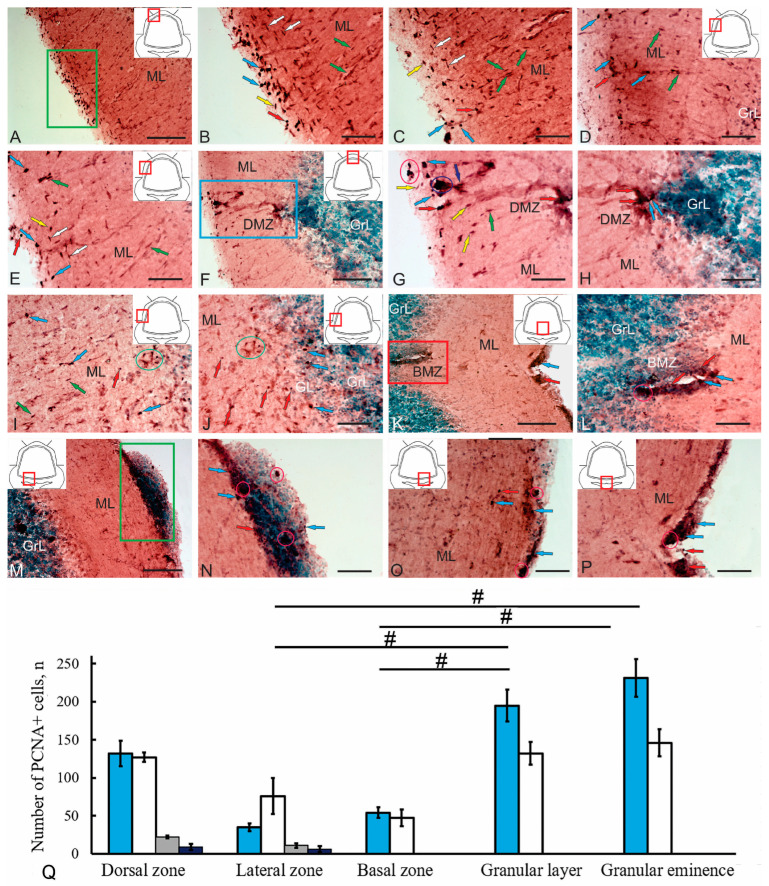
Immunohistochemical labeling of PCNA in the cerebellum of juvenile *Oncorhynchus keta*. (**A**) PCNA immunolocalization in the dorsal region (topography shown in the red square in the pictogram) of the molecular layer (ML); the surface population of PCNA+ cells shown in a green rectangle. (**B**) An enlarged fragment of a green rectangle in (**A**); oval cells (blue arrows), rounded cells (red arrows), intensely labeled cells, tangentially migrating surface cells (yellow arrows), deep cells (white arrows), radially migrating cells (green arrows). (**C**) Localization of PCNA in the dorsal region of the cerebellum, immunolocalization in the upper third of the ML, designations are as in (**B**). (**D**) Localization of PCNA in the dorsolateral region of the cerebellum (topography shown in the pictogram in the red square) in ML; patterns of radial migration to the granular layer (GrL); designations are as in (**B**). (**E**) Localization of PCNA in the dorsolateral region, patterns of tangential migration; designations are as in (**B**). (**F**) A general view of PCNA immunolocalization in the dorsal matrix zone (DMZ), shown in a blue rectangle. (**G**) An enlarged fragment of the dorsal part of the DMZ; the red oval shows an accumulation of adult-type PCNA+ precursors; the blue oval shows an accumulation of PCNA+ neuroepithelial cells (NECs); individual PCNA+ NECs (dark blue arrow); the remaining designations are as in (**B**). (**H**) Enlarged fragment of the ventral part of the DMZ; the red and blue arrows indicate PCNA+ adult-type neural stem/progenitor cells (aNSPCs). (**I**) Patterns of tangential migration in the lateral zone of PCNA+ cells and an accumulation of aNSPCs (in a green oval) in the lower third of the ML, designations are as in (**B**). (**J**) Localization of PCNA+ cells (in green oval) in the lower third of the ML and GrL of the lateral zone of the cerebellum. (**K**) A general view of PCNA immunolocalization in the basal matrix zone (BMZ), shown in the red rectangle, designations are as in (**B**). (**L**) An enlarged fragment of BMZ, an accumulation of aNSPCs shown in the pink oval; designations are as in (**B**). (**M**) An accumulation of PCNA+ cells on the ventral wall of the cerebellum (in the green rectangle). (**N**) A heterogeneous neurogenic cluster at a higher magnification (in pink oval), designations are as in (**B**,**L**). (**O**) Localization of PCNA+ cells and their clusters in the ventrolateral zone of the cerebellum, designations are as in (**B,N**). (**P**) Localization of PCNA+ cells and their clusters in the ventromedial zone, designations are as in (**B**,**L**,**N**). (**Q**) Comparative distribution of PCNA+ cells in various regions of the cerebellum of *Oncorhynchus keta* (M ± SD); significant intergroup differences between the granular layer and the lateral and basal zones, and the granular eminences and the lateral and basal zones indicated by #—(*p* < 0.05); (*n* = 5 in each group); one-way analysis of variance (ANOVA). Blue columns: type I cells; white: type II; gray: type III; black: type IV. Scale: (**A**,**F**,**K**,**M**) 100 µm; (**B**–**E**,**G**–**J**,**L**,**N**–**P**) 50 µm.

**Figure 7 ijms-26-09267-f007:**
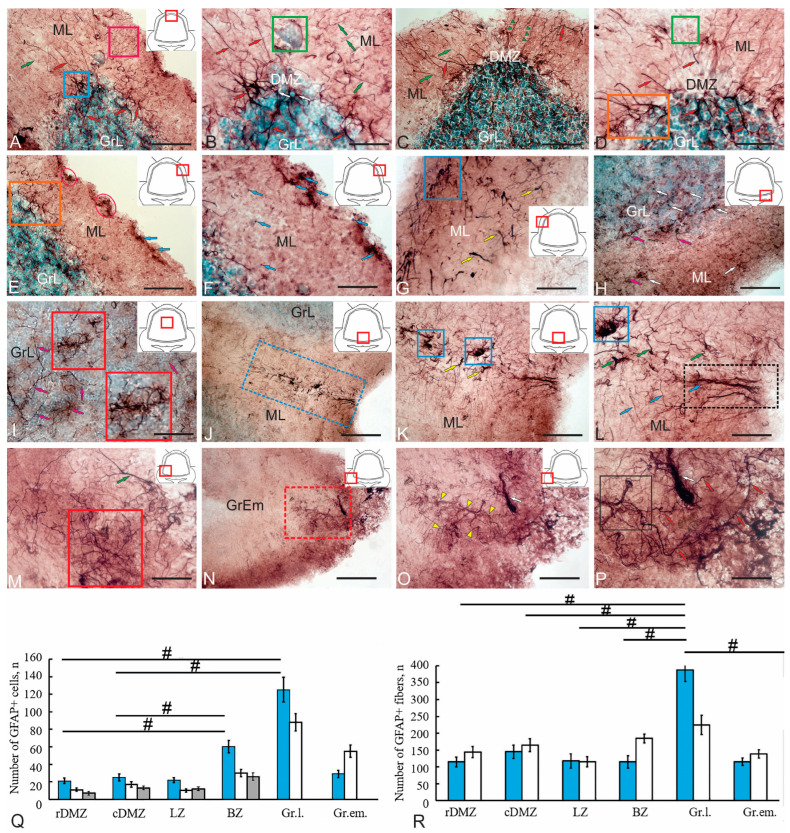
Immunohistochemical labeling of GFAP in the cerebellum of juvenile *Oncorhynchus keta*. (**A**) A general view of GFAP immunolocalization in the rostral part of the dorsal matrix zone (DMZ) (pictogram in the red square); thick radial fibers (red arrows), thin radial fibers (green arrows); clusters of Bergmann glia fibers (in the pink rectangle), thickened plexus of climbing fibers (cf) at the border between molecular layer (ML) and granular layer (GrL) (in the blue rectangle). (**B**) An enlarged fragment of the rostral DMZ; an accumulation of GFAP-negative neuroepithelial cells (NEC) in the green square; GFAP+ cells in the DMZ (white arrows), other designations are as in (**A**). (**C**) A general view of GFAP immunolocalization in the caudal part of the DMZ; bundles of Bergmann glia fibers (green triangular arrows); the remaining designations are as in (**A**,**B**). (**D**) An enlarged fragment of the caudal DMZ, abundant, reticulated branching of GFAP+ fibers at the border of ML and GrL (in the orange rectangle), other designations are as in (**A**). (**E**) Clusters of GFAP+ adult-type neural stem/progenitor cells (aNSPCs) in the lateral subpial region of the cerebellum (in pink ovals); the blue arrows indicate individual GFAP+ aNSPCs; designations are as in (**D**). (**F**) An enlarged fragment showing the distribution of individual GFAP+ aNSPCs (indicated by blue arrows). (**G**) The GFAP+ ends of mossy fibers (mf) (indicated by yellow arrows); a cluster of mf endings in a blue rectangle. (**H**) The individual GFAP+ cells (white arrows) and GFAP+ cf endings (pink arrows) in the basolateral parts of the cerebellum. (**I**) Separate collaterals of GFAP+ cf in the GrL (pink arrows); terminal extensions of mf, and central parts of the mossy rosette (shown in the red inset). (**J**) Distribution patterns of GFAP+ mf (in a blue dotted rectangle) in the mediobasal area of the cerebellum. (**K**) GFAP+ rosettes of mf (blue rectangles) in the mediobasal GFAP+ ascending tract, GFAP+ ends of mf (indicated by yellow arrows). (**L**) A bundle of GFAP+ cf at the base of the cerebellar body (shown in a black dotted rectangle); thin cf (blue arrows), thick cf (green arrows). (**M**) Sockets of GFAP+ mf with numerous filamentous appendages (in a red square) in the basolateral regions of the cerebellum, designations are as in (**L**). (**N**) GrEm of GFAP+ mf formed large rosettes with an uneven surface (in the red dotted rectangle). (**O**) The central part of the GFAP+ rosette formed numerous thin filamentous appendages (indicated by yellow triangular arrows) forming a wide arborization, the glomerulus indicated by a white arrow. (**P**) An enlarged fragment of the central part of the rosette (in a black square); glomerulus (white arrow) and terminal arborizations (indicated by red arrows). (**Q**) Comparative distribution of GFAP+ cells in various regions of the cerebellum of *Oncorhynchus keta* (M ± SD); significant intergroup differences between the granular layer (GrL) and rostral (rDMZ) and caudal parts of the DMZ (cDMZ), basal zone (BZ) and cDMZ, BZ and lateral zone (LZ)—# (*p* < 0.05); (n = 5 in each group), one-way ANOVA. The blue columns represent type I cells, the white ones represent type II cells, and the gray ones represent type III cells. (**R**) Comparative distribution of GFAP+ fibers in various regions of the cerebellum of *Oncorhynchus keta* (mean ± SD); significant intergroup differences between the granular layer (GrL) and the rostral (rDMZ) and caudal parts of the DMZ (cDMZ), lateral (LZ), basal (BZ) zones and granular eminence (GrEm)—# (*p* < 0.05); (n = 5 in each group), one-way analysis of variance (ANOVA). The blue columns represent thin fibers, the white ones represent thick fibers. Scale: (**A**,**C**,**E**,**H**,**K**,**O**) 100 µm; (**B**,**D**,**F**,**G**,**I**,**L**,**M**,**P**) 50 µm; (**J**,**N**) 200 µm.

**Figure 8 ijms-26-09267-f008:**
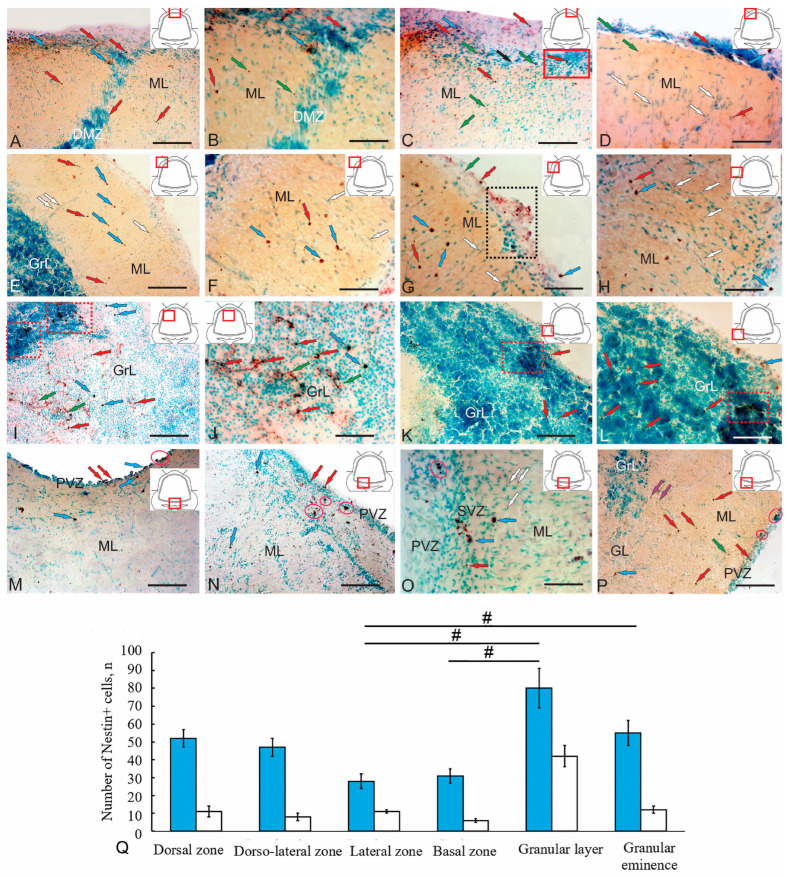
Immunohistochemical labeling of nestin in the cerebellum of juvenile chum salmon *Oncorhynchus keta*. (**A**) Immunolocalization of nestin (Nes) in the dorsal part of the cerebellum (pictogram in the red square); small oval intensely labeled non-glial aNSPCs (blue arrows), round non-glial adult-type neural stem/progenitor cells (aNSPCs) (red arrows). (**B**) An enlarged fragment of the dorsal matrix zone (DMZ), nestin+ granules (green arrows), other designations are as in (**A**). (**C**) Immunolocalization of nestin in the dorsolateral subpial part of the cerebellum (pictogram in the red square); black arrows indicate nestin-negative tangentially migrating cells; an accumulation of nestin-negative cells in the surface layers of the molecular layer (ML) (in the red rectangle); the remaining designations are as in (**A**). (**D**) Patterns of radial migration of nestin-negative cells (white arrows) in the dorsolateral part of the cerebellum. (**E**) Patterns of distribution of nestin+ aNSPCs in the lateral part of the cerebellar body; nestin-negative cells migrating along blood vessels (white arrows); designations are as in (**A**). (**F**) An enlarged fragment of the dorsolateral region showing non-glial nestin+ aNSPCs (blue and red arrows), nestin-negative cells (white arrows). (**G**) A thickening containing clusters of nestin+ granules (in a black dotted rectangle) in the subpial parts of the lateral region; other designations are as in (**D**). (**H**) Patterns of radial migration of nestin-negative cells, designations are as in (**D**). (**I**) A large population of nestin+ cells in the GrL forming clusters (in red dotted rectangles); a heterogeneous population of oval and round nestin+ non-glial precursors (indicated by blue and red arrows); nestin+ granules (green arrows). (**J**) An enlarged fragment showing patterns of extracellular localization of nestin and immunopositive granules (green arrows) adjacent to the DMZ; other designations are as in (**I**). (**K**) Foci of nestin+ type II aNSPCs (in the red dotted rectangle) in the granular eminences. (**L**) An enlarged fragment showing individual nestin+ cells and granules in the GrL of the granular eminences (GrEm), designations are as in I. (**M**) A general view of the paramedian basal zone of the cerebellum; nestin+ aNSPCs of types I and II (blue and red arrows) forming small clusters (in pink ovals) were identified in the periventricular zone (PVZ), bordering the fourth ventricle (IV). (**N**) In the lateral part of the basal zone of the cerebellum, the PVZ included several single and paired nestin+ aNSPCs of type II (red arrows); other designations are as in (**M**). (**O**) An enlarged fragment of PVZ and subventricular zone (SVZ) in the basal part of the cerebellum; nestin-negative tangentially migrating cells indicated by white arrows; other designations are as in (**M**). (**P**) The most lateral areas of the BZ; nestin+ types I and II aNSPCs (blue and red arrows), nestin+ granules (green arrows), PCs (purple arrows); the aNSPC clusters in the PVZ are outlined in red ovals. (**Q**) Comparative distribution of Nes+ cells in various regions of the cerebellum of *O. keta* (M ± SD); significant intergroup differences between the granular layer and the lateral, basal, and granular eminences are indicated by # (*p* < 0.05); (n = 5 in each group); one-way analysis of variance (ANOVA). The blue columns—type I cells, the white ones—type II cells. Scale: (**A**,**E**,**I**,**K**,**M**,**N**,**P**) 100 µm; (**B**–**D**,**F**–**H**,**J**,**L**,**O**) 50 µm.

**Figure 9 ijms-26-09267-f009:**
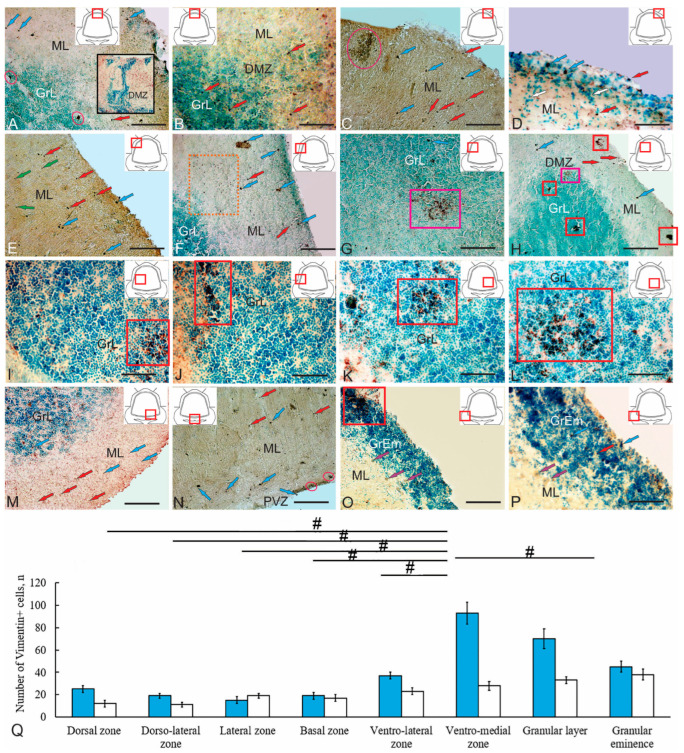
Immunohistochemical labeling of vimentin in the cerebellum of juvenile chum salmon *Oncorhynchus keta*. (**A**) Vimentin (Vim) expression in the dorsal zone of the cerebellum (shown in the pictogram) of juvenile chum salmon in adult-type neural stem/progenitor cells (aNSPCs) of types I and II (blue and red arrows); the dorsal matrix zone (DMZ) is shown in a black rectangle; aNSPC clusters in pink ovals. (**B**) An enlarged fragment of the DMZ, rounded aNSPCs are indicated by red arrows. (**C**) Vimentin+ cluster of aNSPCs in the dorsolateral region (shown in a dark pink oval) types I and II; other designations are as in (**A**). (**D**) An enlarged fragment of the surface zone in the dorsolateral region containing vimentin+ type I aNSPCs (blue arrows) and type II in the upper third of the molecular layer (ML) (red arrows); patterns of radial migration of vimentin-negative cells (white arrows). (**E**) Separate subcellular vimentin+ granules (green arrows) in the dorsolateral areas, other designations are as in (**A**). (**F**) An accumulation of vimentin+ granules (in the orange dotted rectangle) in the lateral zone of the cerebellum, other designations are as in (**A**). (**G**) Sparse clusters of vimentin+ aNSPCs in the dorsal part of the granular layer (GrL) (in the pink rectangle); single vimentin+ aNSPCs of type I (blue arrow). (**H**) Local dense clusters of vimentin+ aNSPCs (in red squares); other designations are as in (**A**,**G**). (**I**) Heterogeneous clusters of vimentin+ aNSPCs; immunopositive neuropil and extracellular deposits of vimentin (in the red rectangle) in the GrL. (**J**) Similar structures (in the red rectangle) at the border of the ML and GrL (see pictogram). (**K**) In the ventrolateral part of the GrL (see pictogram). (**L**) In the ventromedial part of the GrL (see pictogram). (**M**) Distribution of vimentin+ aNSPCs of type I and type II (blue and red arrows) in the ventrolateral part of the basal zone (BZ). (**N**) Distribution of vimentin+ aNSPCs of type I and type II in the ventromedial part of the BZ (blue and red arrows), pink ovals show clusters of vimentin+ aNSPCs in the periventricular zone (PVZ) of the IV ventricle. (**O**) Local neurogenic niche with vimentin+ aNSPCs (in the red rectangle) in the granular eminence (GrEm), vimentin-negative ML (purple arrows). (**P**) An enlarged GrEm fragment showing vimentin+ types I and II aNSPCs (blue and red arrows); vimentin–negative ML (purple arrows). (**Q**) Comparative distribution of Vim+ cells in various regions of the cerebellum of *Oncorhynchus keta* (M ± SD); significant intergroup differences between the ventromedial region of the granular layer and the granular layer, dorsal, dorsolateral, lateral, basal and ventrolateral zones are indicated by # (*p* < 0.05); (n = 5 in each group); one-way analysis of variance (ANOVA). The blue columns represent type I cells, the white ones represent type 2 cells. Scale: (**A**,**C**,**E**–**H**,**M**–**O**) 100 µm; (**B**,**D**,**I**–**L**,**P**) 50 µm.

**Figure 10 ijms-26-09267-f010:**
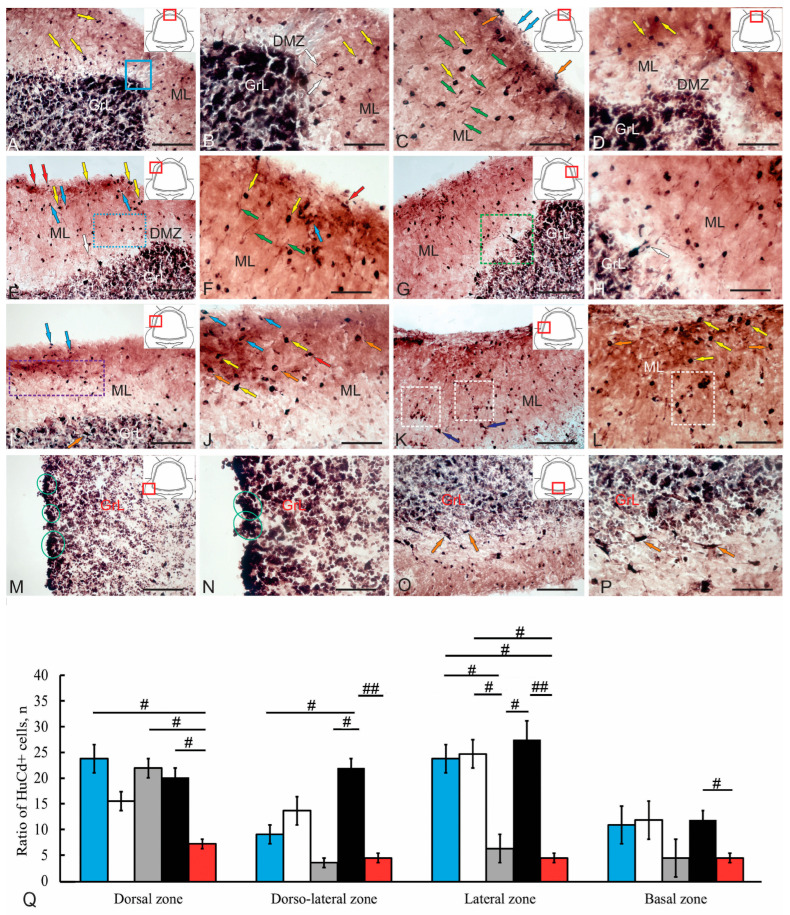
Immunohistochemical labeling of HuCD in the cerebellum of juvenile chum salmon *Oncorhynchus keta*. (**A**) The localization of HuCD in the dorsal part of the cerebellum (pictogram), the HuCD-negative dorsal matrix zone (DMZ) is shown in a blue rectangle, paired HuCD+ neurons are indicated by yellow arrows. (**B**) An enlarged fragment of the DMZ, HuCD+ cells are localized at the border of the granular layer (GrL) and molecular layer (ML), eurydendroid cells (EDCs) with axonal arborization (white arrows), undifferentiated HuCD+ cells in the ML are indicated by yellow arrows. (**C**) A fragment of the dorsolateral part of the ML containing intensely labeled HuCD+ cells (yellow arrows), superficially located HuCD+ cells with an immunonegative nucleus (orange arrows), HuCD-negative adult-type neural stem/progenitor cells (aNSPCs) (blue arrows), HuCD-negative radially migrating cells (green arrows). (**D**) The dorsal region adjacent to the DMZ, containing HuCD+ cells (yellow arrows). (**E**) HuCD+ cells in the dorsal part of the cerebellum: superficial small undifferentiated cells (red arrows), oval single cells or forming small clusters of cells (yellow arrows), small moderately labeled bipolar cells (blue arrows); HuCD+ neurons forming connections in the ML (in the blue dotted rectangle); axonal afferent HuCD+ connections, at the border of the ML and GL (white arrow). (**F**) Morphogenetic patterns of HuCD expression in cells of various types, designations are as in (**C**,**E**). (**G**) Mass labeling of HuCD in cells of the surface layers of the dorsolateral zone of ML; in the green dotted rectangle HuCD+ EDCs are indicated. (**H**) An enlarged fragment in the green dotted rectangle in (**G**), the EDC axon with varicose microcytosculpture is indicated by a white arrow. (**I**) Heterogeneous HuCD+ cells in the lateral part of the superficial and middle layers of the ML; the purple dotted rectangle shows an accumulation of immunopositive cells in the middle part of the ML; the orange arrow indicates the EDC, the blue arrows indicate the surface of an undifferentiated HuCD+ cell. (**J**) An enlarged fragment in the purple rectangle in I, designations are as in (**C**,**E**). (**K**) An accumulation of HuCD+ cells in the deep layers of the ML; forming foci of neuronal differentiation (in white dotted rectangles) in the lateral zone; the dark blue arrows indicate developing projection neurons. (**L**) An enlarged fragment of the ML, HuCD+ cells with an immunonegative nucleus (orange arrows), other designations are as in (**E**,**K**). (**M**) Small HuCD+ cells and their clusters (in green ovals) in the surface layers of granular eminences (GrEm). (**N**) An enlarged fragment of HuCD+ clusters in the surface layers of GrEm. (**O**) Immunolocalization of HuCD in the ventromedial part of the BZ, EDC is indicated by orange arrows. (**P**) An enlarged fragment showing the morphology of HuCD+ EDCs (orange arrows). (**Q**) Comparative distribution of HuCD+ cells in various regions of the cerebellum of *O. keta* (M ± SD); significant intergroup differences between the granular layer, dorsal, dorsolateral, lateral and basal zones are indicated by # (*p* < 0.05); ## (*p* < 0.01); (n = 5 in each group); one-way analysis of variance (ANOVA). The blue columns represent type I cells, the white ones—type II, the gray ones—type III, the black ones—type IV, the red ones—type V, and the yellow ones—type VI. Scale: (**A**,**E**,**G**,**I**,**K**,**M**,**O**) 100 µm; (**B**–**C**,**F**,**H**,**J**,**L**,**N**,**P**) 50 µm.

**Figure 11 ijms-26-09267-f011:**
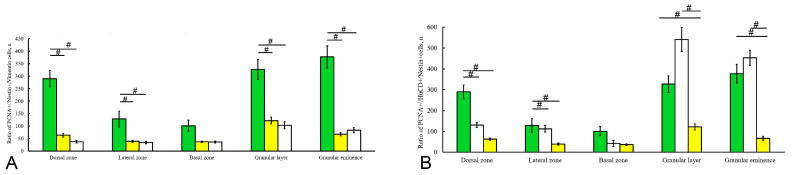
Comparative distribution of immunohistochemical markers in the cerebellum of juvenile chum salmon *Oncorhynchus keta*. (**A**) Comparative distribution of PCNA+, Nes+ and Vim+ cells in different regions of the cerebellum (M ± SD); significant intergroup differences are indicated by # (*p* < 0.05); (n = 5 in each group); one-way analysis of variance (ANOVA). The green columns represent PCNA+ cells, the yellow columns represent Nes+ cells, and the white columns—Vim+ cells. (**B**) Comparative distribution of PCNA+, HuCD+, Nes+ cells in various regions of the cerebellum (M ± SD); significant intergroup differences are indicated by # (*p* < 0.05); (n = 5 in each group); one-way analysis of variance (ANOVA). The green columns represent PCNA+ cells, the white columns—Vim+ cells, and the yellow columns—Nes+ cells.

**Table 1 ijms-26-09267-t001:** Ultrastructural characteristics of cerebellum in juvenile chum salmon, *O. keta*.

Cells of Molecular Layer				
Type of Cells	Stellate Cells		Non-glial aNSPCs (DMZ)	
Long axis of cells soma (µm)	8.47 ± 0.66		7.71 ± 0.45	
Short axis of cells soma (µm)	7.48 ± 0.78		6.06 ± 0.53	
Sample size	*N* = 22		*N* = 18	
**Nucleus**				
Contour	Round or oval		Round, smooth	
Long axis (µm)	6.79 ± 0.76		6.65 ± 0.36	
Short axis (µm)	5.02 ± 0.21		5.46 ± 0.54	
Chromatin	Denser euchromatin and heterochromatin		reticular euchromatin	
Color	medium		light	
Nucleoli	1–2 are rarely found		No or 1	
**Cytoplasm**				
Percentage/color	medium		light	
Mitochondria	many		few	
Vacuoles	some		no	
Lipid droplets	Yes		no	
Dense bodies	No		no	
Cell contacts	No		Non-glial aNSPCs, glial aNSPCs III	
**glial aNSPCs (GrL, ML)**				
**Type of Cells**	**III**	**IV**		**DMZ**
Long axis of cells soma (µm)	5.55 ± 0.6	5.67 ± 1.13		5.32 ± 0.92
Short axis of cells soma (µm)	3.87 ± 0.67	3.81 ± 0.42		3.07 ± 0.66
Sample size	*N* = 15	*N* = 15		*N* = 20
**Nucleus**				
Contour	elongated; irregular ± invaginations	ovoid, irregular ± invaginations		irregular ± invaginations
Long axis (µm)	4.81 ± 0.48	4.87 ± 1.21		4.73 ± 0.9
Short axis (µm)	3.39 ± 0.63	3.39 ± 0.35		2.62 ± 0.67
Chromatin	evenly distributed; non-clumped	reticulated; clumped hetero		reticulated; clumped hetero
Color	medium	dark		dark
Nucleoli	1–2	1 or 2, rarely visible		No visible
**Cytoplasm**				
Percentage/color	scanty, dark	scanty, medium		scanty, medium
Mitochondria	few	few		single
Vacuoles	no	several, rarely encountered		no
Lipid droplets	0–1	no		no
Dense bodies	yes	no		no
**Localization**	ML, GrL	ML, GL, GrL		ML
**Cell contacts**	III, IV	III, IV		III, NEC

**Table 2 ijms-26-09267-t002:** Morphometric and densitometric characteristics of PCNA-labeled cells (M ± SD) in the intact cerebellum of juvenile chum salmon, *O. keta.*

Brain Areas	Type of Cells	Cell Size, µm *	Optical Density **, UOD
**Dorsal zone**	Oval	5.2 ± 0.8/3.4 ± 0.6	+++
	Oval	7.0 ± 0.6/5.4 ± 0.8	+++
	Oval	7.4 ± 0.9/3.2 ± 0.4	++/+++
	Elongated	10.9 ± 1.1/3.6 ± 0.5	++/+++
	Elongated	16.4 ± 1.1/3.1 ± 0.3	++
**Lateral zone**	Oval	4.7 ± 0.5/3.5 ± 0.5	+++
	Oval	7.1 ± 0.7/3.6 ± 0.7	++/+++
	Elongated	8.5 ± 0.8/3.3 ± 0.6	+++
	Elongated	15.5 ± 1.3/3 ± 0.3	++/+++
**Basal zone**	Oval	4.4 ± 0.7/3.4 ± 0.6	++/+++
	Oval	6.9 ± 0.8/3.9 ± 0.6	++/+++
**Granular layer**	Round	4.5 ± 0.1/3.7 ± 0.6	++/+++
	Oval	6.1 ± 0.5/4.3 ± 0.4	++/+++
**Granular eminence**	Round	5.3 ± 0.5/3.6 ± 0.7	++/+++
	Oval	6.2 ± 0.2/4.2 ± 0.8	+++

* The values before and after slash (/) are for large and small diameters of the cell body. ** Optical density (OD) in cells was classified according to the following scale: high (150–110 UOD, which corresponds to +++) and moderate (110–70 UOD, which corresponds to ++).

**Table 3 ijms-26-09267-t003:** Morphometric and densitometric characteristics of GFAP-labeled cells (M ± SD) in the intact cerebellum of juvenile chum salmon, *O. keta.*

Brain Areas	Type of Cells	Cell Size, µm *	Optical Density **, UOD
**Dorsal zone rostral**	Round	4.0 ± 0.4/3.4 ± 0.5	+++
	Round	4.9 ± 0.3/3.8 ± 0.4	+++
	Round	6.5 ± 0.7/4.5 ± 0.5	+++
		**Fibers**	
		143.6 ± 14.2/1.7 ± 0.4	++/+++
		158.4 ± 16.1/2.5 ± 0.2	+++
**Dorsal zone caudal**	Round	4.0 ± 0.4/3.4 ± 0.5	+++
	Round	4.9 ± 0.3/3.8 ± 0.4	+++
	Round	6.5 ± 0.7/4.5 ± 0.5	+++
		**Fibers**	
		123.6 ± 34.2/1.6 ± 0.3	++/+++
		167.1 ± 10.9/2.5 ± 0.2	+++
**Lateral zone**	Round	6.5 ± 0.4/4.8 ± 0.2	++/+++
	Round	7.5 ± 0.7/5.7 ± 0.5	+++
	Round	8.7 ± 0.7/6.9 ± 0.4	+++
		**Fibers**	
		112.4 ± 14.1/1.7 ± 0.2	++/+++
		157 ± 14.4/2.6 ± 0.3	+++
**Basal zone**	Round	3.5 ± 0.3/3.0 ± 0.3	++/+++
	Round	4.5 ± 0.3/3.7 ± 0.5	+++
	Round	5.7 ± 0.4/4.4 ± 0.9	+++
		**Fibers**	
		95.5 ± 34.6/2.1 ± 0.3	++/+++
		109.0 ± 28.6/1.3 ± 0.3	+++
**Granular layer**	Round	3.8 ± 0.5/3 ± 0.3	++/+++
	Round	5.4 ± 0.6/3.1 ± 0.3	+++
		**Fibers**	
		133.7 ± 24.1/1.7± 0.4	++/+++
		147.4 ± 16.3/2.2 ± 0.4	+++
**Granular eminence**	Round	5.4 ± 0.1/3.9 ± 0.8	++/+++
	Round	3.7 ± 0.5/3.4 ± 0.6	+++
		**Fibers**	
		111.5 ± 18.3/1.8 ± 0.4	++/+++
		151.2 ± 59.9/2.6 ± 0.2	+++

* The values before and after slash (/) are for large and small diameters of the cell body. ** Optical density (OD) in cells was classified according to the following scale: high (130–100 UOD, which corresponds to +++) and moderate (100–70 UOD, which corresponds to ++).

**Table 4 ijms-26-09267-t004:** Morphometric and densitometric characteristics of nestin-labeled cells (M ± SD) in the intact cerebellum of juvenile chum salmon, *O. keta.*

Brain Areas	Type of Cells	Cell Size, µm *	Optical Density **, UOD
**Dorsal zone**	Granules	3 ± 0.3/2.6 ± 0.4	++/+++
	Round	4.4 ± 0.5/3.7 ± 0.4	++/+++
	Oval	6.8 ± 0.6/5.2 ± 0.6	+++
**Dorso-lateral zone**	Granules	2.9 ± 0.2/2.6 ± 0.2	++
	Round	4.4 ± 0.2/3.3 ± 0.5	++/+++
	Oval	6.4 ± 0.5/5.7 ± 0.5	++/+++
**Lateral zone**	Granules	2.8 ± 0.3/2.7 ± 0.2	++
	Round	4.1 ± 0.4/3.3 ± 0.4	++/+++
	Oval	6.7 ± 0.9/4.8 ± 0.4	++/+++
**Basal zone**	Granules	2.9 ± 0.2/2.7 ± 0.2	++/+++
	Round	4.2 ± 0.5/3.7 ± 0.4	++/+++
	Oval	6.4 ± 0.3/5.6 ± 0.5	+++
**Granular layer**	Granules	3.1 ± 0.2/2.6 ± 0.2	++/+++
	Round	3.6 ± 0.2/3.2 ± 0.3	+++
	Round	6.4 ± 0.3/4.9 ± 0.5	++/+++
**Granular eminence**	Granules	3 ± 0.3/2.6 ± 0.3	++/+++
	Round	3.9 ± 0.5/3.5 ± 0.5	++/+++
	Oval	6.5 ± 0.4/4.6 ± 0.7	++/+++

* The values before and after slash (/) are for large and small diameters of the cell body. ** Optical density (OD) in cells was classified according to the following scale: high (180–120 UOD, which corresponds to +++), moderate (120–80 UOD, which corresponds to ++).

**Table 5 ijms-26-09267-t005:** Morphometric and densitometric characteristics of vimentin-labeled cells (M ± SD) in the intact cerebellum of juvenile chum salmon, *O. keta*.

Brain Areas	Type of Cells	Cell Size, µm *	Optical Density **, UOD
**Dorsal zone**	Granules	3.1 ± 0.3/2.9 ± 0.3	++/+++
	Round	4.5 ± 0.3/3.5 ± 0.5	+++
	Oval	5.8 ± 0.4/3.8 ± 0.5	+++
**Dorso-lateral zone**	Granules	3 ± 0.3/2.8 ± 0.3	++
	Round	4.3 ± 0.4/3.4 ± 0.3	+++
	Oval	5.6 ± 0.5/3.9 ± 0.5	++/+++
**Lateral zone**	Granules	3 ± 0.2/2.4 ± 0.2	++/+++
	Round	4.4 ± 0.4/3.6 ± 0.2	++/+++
	Oval	5.2 ± 0.2/3.6 ± 0.5	+++
**Ventro-lateral zone**	Granules	2.9 ± 0.1/2.5 ± 0.3	++/+++
	Round	4.3 ± 0.4/3.4 ± 0.4	+++
	Oval	5.6 ± 0.3/3.4 ± 0.4	+++
**Basal zone**	Granules	2.8 ± 0.3/2.6 ± 0.4	++/+++
	Round	4.2 ± 0.3/3.4 ± 0.5	+++
	Oval	5.5 ± 0.3/3.8 ± 0.5	+++
**Ventro-medial zone**	Granules	3 ± 0.3/2.7 ± 0.4	++/+++
	Round	4.4 ± 0.3/3.5 ± 0.4	+++
	Oval	5.6 ± 0.4/3.7 ± 0.3	+++
**Granular layer**	Granules	2.8 ± 0.3/2.6 ± 0.3	++
	Round	4.3 ± 0.3/3.6 ± 0.5	+++
	Oval	6.2 ± 0.6/3.5 ± 0.5	++/+++
**Granular eminence**	Granules	2.7 ± 0.3/2.8 ± 0.4	++/+++
	Round	4.3 ± 0.2/3.5± 0.4	+++
	Oval	5.2 ± 0.6/3.2 ± 0.2	+++

* The values before and after slash (/) are for large and small diameters of the cell body; ** Optical density (OD) in cells was classified according to the following scale: high (160–120 UOD, which corresponds to +++), moderate (120–80 UOD, which corresponds to ++).

**Table 6 ijms-26-09267-t006:** Morphometric and densitometric characteristics of HuCd-labeled cells (M ± SD) in the intact cerebellum of juvenile chum salmon, *O. keta*.

Brain Areas	Type of Cells	Cell Size, µm *	Optical Density **, UOD
**Dorsal zone**	Round	6.9 ± 0.5/6.6 ± 0.6	+
	Oval	13.7 ± 1.4/11.7 ± 1	++/+++
	Bipolar	11.3 ± 1.4/6.5 ± 0.6	++/+++
	Oval	13.4 ± 1.1/11.2 ± 1	+++
	EDC	27.5 ± 1.3/15.7 ± 1.5	+++
**Dorso-lateral zone**	Round	7.9 ± 0.7/7.6 ± 0.6	+
	Oval	14.2 ± 1.2/11.3 ± 1	++/+++
	Bipolar	11.7 ± 1.2/6.7 ± 0.6	++/+++
	Oval	15.2 ± 1.3/10.3 ± 1.2	++/+++
	EDC	25.4 ± 2.3/14.6 ± 1.8	+++
**Lateral zone**	Round	8.5 ± 0.5/7.3 ± 0.6	+
	Oval	15.4 ± 1.2/11.5 ± 0.9	++/+++
	Bipolar	13.4 ± 1.4/7.1 ± 0.6	++/+++
	Oval	16.8 ± 1.1/10.2 ± 1.4	+++
	EDC	28.3 ± 1.3/13.5 ± 1.7	+++
**Basal zone**	Round	8.6 ± 0.5/8.4 ± 0.6	+
	Oval	12.5 ± 1/10.8 ± 0.9	++/+++
	Oval	14.9 ± 1.1/9.3 ± 1.1	+++
	EDC	29.6 ± 2.8/15.3 ± 1.4	+++
**Granular layer**	Oval	4.1 ± 0.3/3.2 ± 0.4	++/+++
	Round	5.9 ± 0.6/5.1 ± 0.6	++/+++
**Granular eminence**	Oval	5.3 ± 0.5/4.6 ± 0.6	++/+++
	Oval	6.7 ± 0.6/5.3 ± 0.8	++/+++

* The values before and after slash (/) are for large and small diameters of the cell body. ** Optical density (OD) in cells was classified according to the following scale: high (200–150 UOD, which corresponds to +++), moderate (150–100 UOD, which corresponds to ++) and less than 100 UOD, weak (corresponds to +).

**Table 7 ijms-26-09267-t007:** Characteristics of primary antibodies used in immunohistochemical studies.

No.	Antibodies	Manufacturer	Dilution	Catalog Number	Marker
1	GFAP	Abcam, Cambridge CB2 0AX, UK.	1:300	GF5 Catalog No. ab10062	GFAP
2	PCNA	Novus Biologicals, Centennial, CO, USA	1:300	Catalog No. NB500-106, Lot A2,	PCNA
3	Vimentin	Abcam, Cambridge CB2 0AX, UK.	1:300	Catalog No. ab28028	Vimentin
4	Nestin	Abcam, Cambridge CB2 0AX, UK.	1:300	clone 2C1.3A11; Catalog No. ab18102	Nestin
5	HuCD	Invitrogen™, Thermo Fisher Scientific, Waltham, MA, USA.	1:300	clone 16A11; Catalog No. A21271;	HuCD

## Data Availability

The original contributions presented in this study are included in the article/Appendix A. Further inquiries can be directed to the corresponding author.

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
