# Peer review of "Immunohistochemical and Ultrastructural Analysis of Adult Neurogenesis Involving Glial and Non-Glial Progenitors in the Cerebellum of Juvenile Chum Salmon Oncorhynchus keta"

_ijms, 2025, doi:10.3390/ijms26199267_

Round 1

Reviewer 1 Report

Comments and Suggestions for Authors

The manuscript “Immunohistochemical and Ultrastructural Analysis of Adult Neurogenesis Involving Glial and Non-Glial Progenitors in the Cerebellum of Juvenile Chum Salmon Oncorhynchus keta” by Pushchina et al. is a detailed description of the ultrastructural organization of various cell types involved in homeostatic growth in the cerebellum of juvenile chum salmon, with additional immunohistochemical verification of stem, proliferating and differentiated cells. This is a significant work that provides a certain research basis for future study. Overall, the manuscript reads well, the only drawback being the excessive volume of text.

I recommend that authors critically reread the manuscript and shorten the text a little without compromising the quality. For example, elements of discussion can be removed from the Results section, etc.

The ultrastructure drawings are of excellent quality. However, there are some problems with the presentation of the illustrations showing the results of immunocytochemical labeling. The close up is too small. To my mind that need improvement.

Minor point:

line 84-86. The cells do not express bromodeoxyuridine (BrdU). BrdU may be incorporated into DNA during incubation experiments. Please correct the text.

Author Response

The manuscript “Immunohistochemical and Ultrastructural Analysis of Adult Neurogenesis Involving Glial and Non-Glial Progenitors in the Cerebellum of Juvenile Chum Salmon Oncorhynchus keta” by Pushchina et al. is a detailed description of the ultrastructural organization of various cell types involved in homeostatic growth in the cerebellum of juvenile chum salmon, with additional immunohistochemical verification of stem, proliferating and differentiated cells. This is a significant work that provides a certain research basis for future study. Overall, the manuscript reads well, the only drawback being the excessive volume of text.

I recommend that authors critically reread the manuscript and shorten the text a little without compromising the quality. For example, elements of discussion can be removed from the Results section, etc.

Thank you for this recommendation. The Results section has been carefully proofread and recommended abbreviations have been added. However, the volume of reductions was limited by the need to add some additional data regarding the IHC phenotypes of cells of types I and II identified during ultrastructural analysis, as well as the Limitations section.

The ultrastructure drawings are of excellent quality. However, there are some problems with the presentation of the illustrations showing the results of immunocytochemical labeling. The close up is too small. To my mind that need improvement.

Thank you for your comment. In accordance with the recommendations, separate fragments of complex installations have been prepared, illustrating the close-up (at x40 magnification) in Supplementary materials. The corresponding clarifications have been added to the text of the results.

Minor point:

line 84-86. The cells do not express bromodeoxyuridine (BrdU). BrdU may be incorporated into DNA during incubation experiments. Please correct the text.

Thank you for your comment. Appropriate corrections have been made to the manuscript.

Reviewer 2 Report

Comments and Suggestions for Authors

The manuscript entitle “Immunohistochemical and Ultrastructural Analysis of Adult 2 Neurogenesis Involving Glial and Non-Glial Progenitors in the 3 Cerebellum of Juvenile Chum Salmon Oncorhynchus keta” presents an extensive, well defined and presented immunohistochemical and ultrastructural analysis of adult neurogenesis within the cerebellum of juvenile chum salmon (Oncorhynchus keta). The study is strengthened by its comprehensive, multimodal methodology. The work delivers significant novel insights, notably providing the first ultrastructural evidence of eurydendroid cells (EDCs) in this species and identifying distinct populations of glial and non-glial adult-type neural stem/progenitor cells with differential migratory behaviors. Furthermore, the authors attractively contextualize their discoveries within the broader field through strong comparative analysis with zebrafish and trout data, underscoring important aspects of phylogenetic conservation in cerebellar neurogenic mechanisms. While the following revisions are recommended to enhance clarity and statistical reporting, the core findings are novel, well-supported by the data, and presented with commendable rigor.

Major Comments

• A notable point requiring clarification from the authors pertains to the terminology and classification employed for 'adult-type neural stem/progenitor cells (aNSPCs)'. The current usage of this term appears ambiguous within the manuscript. Specifically, it is recommended that the authors explicitly define whether the referenced type III/IV cells (according to the Lindsey classification) correspond to quiescent neural stem cells or transit-amplifying progenitor cells. Resolving this terminological precision is crucial for accurately interpreting the cellular populations under investigation.

• A significant gap persists in the ultrastructural characterization concerning the molecular identity of the described dark cells (type I) and type II glial cells. While their distinctive morphological features are presented, these cell populations are not currently defined by definitive molecular or phenotypic markers within the manuscript, hindering their precise classification and functional interpretation within the proposed framework. If the authors possess data linking these ultrastructural types to established molecular markers (such as GFAP, vimentin, or nestin via immuno-EM or other methods), it would be highly valuable to include this information, potentially within supplementary files. Such data would significantly strengthen the correlation between morphology and phenotype. 

• One other limitation concern in my point of view is the reliance on correlative markers (PCNA+ for proliferation, HuCD+ for differentiation) to infer cellular behavior without accompanying functional validation. While these indicators provide valuable initial insights, they inherently lack the capacity to establish causal relationships or trace definitive lineage progression. It would be valuable to clarify whether the authors explicitly acknowledged this inherent limitation of marker-based interpretation within their discussion section, as recognizing this constraint is crucial for contextualizing the findings.

• A main limitation in statistical reporting concerns the incomplete presentation of ANOVA results, as exemplified in Figure 6Q. While statistical significance is indicated, key quantitative metrics, specifically effect sizes (e.g., η² [eta-squared] or Cohen’s d) and details of post-hoc correction methods are not fully reported. Effect sizes are essential for contextualizing the biological relevance of observed differences beyond mere statistical significance. Additionally, the specific post-hoc methodology applied (e.g., Bonferroni vs. Dunnett’s correction) should be explicitly stated to ensure transparency in multiple comparison adjustments. Including these elements would significantly strengthen the analytical rigor and facilitate more robust interpretation of the quantitative findings

Minor Comments

• Page 5, line 135: "GrL and GrEm" → Specify granular eminences (GrEm) once in full. • Figure 1F caption: Clarify "asymmetric excitatory type synapses" with arrowheads. • Section 3.2: "GFAP+afferents undergo development" → Rephrase to "GFAP+afferentsshow dorsoventral maturation gradients."

Author Response

The manuscript entitle “Immunohistochemical and Ultrastructural Analysis of Adult 2 Neurogenesis Involving Glial and Non-Glial Progenitors in the 3 Cerebellum of Juvenile Chum Salmon Oncorhynchus keta” presents an extensive, well defined and presented immunohistochemical and ultrastructural analysis of adult neurogenesis within the cerebellum of juvenile chum salmon (Oncorhynchus keta). The study is strengthened by its comprehensive, multimodal methodology. The work delivers significant novel insights, notably providing the first ultrastructural evidence of eurydendroid cells (EDCs) in this species and identifying distinct populations of glial and non-glial adult-type neural stem/progenitor cells with differential migratory behaviors. Furthermore, the authors attractively contextualize their discoveries within the broader field through strong comparative analysis with zebrafish and trout data, underscoring important aspects of phylogenetic conservation in cerebellar neurogenic mechanisms. While the following revisions are recommended to enhance clarity and statistical reporting, the core findings are novel, well-supported by the data, and presented with commendable rigor.

Thank you for the positive assessment of our work and the special attention that the distinguished reviewer paid to the analysis of the article. It was very important and interesting for us to receive a qualified expert assessment, which in many ways allowed us to better understand the results of the study and give a deeper functional interpretation of the data obtained. Thank you for the deep and interesting recommendations made by the distinguished reviewer, which allowed us to place important accents, thanks to which a more comprehensive study of the results obtained and a more specific conclusion are possible.

Major Comments

  • A notable point requiring clarification from the authors pertains to the terminology and classification employed for 'adult-type neural stem/progenitor cells (aNSPCs)'. The current usage of this term appears ambiguous within the manuscript. Specifically, it is recommended that the authors explicitly define whether the referenced type III/IV cells (according to the Lindsey classification) correspond to quiescent neural stem cells or transit-amplifying progenitor cells. Resolving this terminological precision is crucial for accurately interpreting the cellular populations under investigation.

 We thank Reviewer for this recommendation. Of course, clarifying the proliferative status of type III and IV cells and defining them as resting or transient-enhancing progenitor cells is an important characteristic that provides grounds for further in-depth analysis and functional verification. Appropriate clarifications were made to the Discussion section, according to which pronounced clustering is observed among cells of types III and IV, which is most likely a consequence of their proliferative activity. Analysis of proliferating PCNA+ cells indicates the presence of cells with similar morphological characteristics and a high level of PCNA immunopositivity, which additionally indicates that such cells belong to a transient-enhancing population of progenitor cells.

  • A significant gap persists in the ultrastructural characterization concerning the molecular identity of the described dark cells (type I) and type II glial cells. While their distinctive morphological features are presented, these cell populations are not currently defined by definitive molecular or phenotypic markers within the manuscript, hindering their precise classification and functional interpretation within the proposed framework. If the authors possess data linking these ultrastructural types to established molecular markers (such as GFAP, vimentin, or nestin via immuno-EM or other methods), it would be highly valuable to include this information, potentially within supplementary files. Such data would significantly strengthen the correlation between morphology and phenotype. 

We thank Reviewer for this comment. As part of the immunohistochemical analysis, it was found that type I cells (dark cells) correspond morphologically to GFAP+ small single cells. Such cellular precursors most likely correspond to silent precursors. Type II cells are also GFAP+ cells, but they have large com sizes (Table 2, round II). Type II cells are prone to cluster formation and are PCNA+, which confirms their transient-enhancing phenotype.

  • One other limitation concern in my point of view is the reliance on correlative markers (PCNA+ for proliferation, HuCD+ for differentiation) to infer cellular behavior without accompanying functional validation. While these indicators provide valuable initial insights, they inherently lack the capacity to establish causal relationships or trace definitive lineage progression. It would be valuable to clarify whether the authors explicitly acknowledged this inherent limitation of marker-based interpretation within their discussion section, as recognizing this constraint is crucial for contextualizing the findings.

 Thank you for this comment, of course, the dependence of correlative markers for determining cell behavior without subsequent functional experiments is only limited. We fully recognize this limitation and discuss these issues in a special section.

  • A main limitation in statistical reporting concerns the incomplete presentation of ANOVA results, as exemplified in Figure 6Q. While statistical significance is indicated, key quantitative metrics, specifically effect sizes (e.g., η² [eta-squared] or Cohen’s d) and details of post-hoc correction methods are not fully reported. Effect sizes are essential for contextualizing the biological relevance of observed differences beyond mere statistical significance. Additionally, the specific post-hoc methodology applied (e.g., Bonferroni vs. Dunnett’s correction) should be explicitly stated to ensure transparency in multiple comparison adjustments. Including these elements would significantly strengthen the analytical rigor and facilitate more robust interpretation of the quantitative findings

 Thank you for this valuable comment. The main purpose of the one-sided variance analysis was to compare the cross-group and, in the case of HuCD, intra-group variances. The null hypothesis assumed that there were no differences between the variances, both at the intergroup and intragroup levels. The results of the one-sided ANOVA variance analysis showed the existence of significant differences (p <0.05) in the intergroup variance for PCNA, GFAP, vimentin and nestin and the intragroup variance for HuCD. The Bonferroni correction was used because small groups (n=5) were analyzed. Since we used the built-in Statistica 12 software package, the effect sizes (for example, n2 [eta-squared] or Cohen's coefficient) were not specifically calculated. Perhaps in future studies we will apply this sophisticated algorithm to obtain more complete statistical data.

Minor Comments

  • Page 5, line 135: "GrL and GrEm" → Specify granular eminences (GrEm) once in full.

Corrections have been made

  • Figure 1F caption: Clarify "asymmetric excitatory type synapses" with arrowheads.

Corrections have been made

  • Section 3.2: "GFAP+afferents undergo development" → Rephrase to "GFAP+afferentsshow dorsoventral maturation gradients."

Corrections have been made